# Flow driven robotic navigation of microengineered endovascular probes

Lucio Pancaldi [1], Pietro Dirix[1], Adele Fanelli[2], Augusto Martins Lima [3], Nikolaos Stergiopulos[3], Pascal John Mosimann[4,5], Diego Ghezzi [2] & Mahmut Selman Sakar [1,3✉]

Minimally invasive medical procedures, such as endovascular catheterization, have considerably reduced procedure time and associated complications. However, many regions inside the body, such as in the brain vasculature, still remain inaccessible due to the lack of appropriate guidance technologies. Here, experimentally and through numerical simulations, we show that tethered ultra-flexible endovascular microscopic probes can be transported through tortuous vascular networks with minimal external intervention by harnessing hydrokinetic energy. Dynamic steering at bifurcations is performed by deformation of the probe head using magnetic actuation. We developed an endovascular microrobotic toolkit with a cross-sectional area that is orders of magnitude smaller than the smallest catheter currently available. Our technology has the potential to improve state-of-the-art practices as it enhances the reachability, reduces the risk of iatrogenic damage, significantly increases the speed of robot-assisted interventions, and enables the deployment of multiple leads simultaneously through a standard needle injection and saline perfusion.

[1] Institute of Mechanical Engineering, Ecole Polytechnique Fédérale de Lausanne (EPFL), 1015 Lausanne, Switzerland. [2] Medtronic Chair in Neuroengineering, Center for Neuroprosthetics and Institute of Bioengineering, School of Engineering, EPFL, 1202 Geneva, Switzerland. [3] Institute of Bioengineering, EPFL, 1015 Lausanne, Switzerland. [4] Institute for Diagnostic and Interventional Neuroradiology, 3010 Bern, Switzerland. [5] Department of Diagnostic and Interventional Neuroradiology, Alfried Krupp Krankenhaus, 45130 Essen, Germany. ✉email: selman.sakar@epfl.ch

The cardiovascular system oxygenates the entire body through an exquisitely interconnected fluid network, potentially providing physicians and scientists minimally invasive access to any target tissue. Taking advantage of this opportunity, the diagnosis and treatment of prominent abnormalities and diseases in the brain such as tumours, aneurysms, stroke, and arteriovenous malformations have been routinely accomplished using catheterization techniques[1–5]. Recent work has also demonstrated long-term recording and stimulation of brain dynamics using endovascular stent electrode arrays[6,7]. However, the majority of the brain is still inaccessible because the existing tools are bulky, and navigating through the minuscule and tortuous cerebral vasculature without causing tissue damage is extremely challenging. While technological progress in microengineering has introduced a variety of microdevices that can perform optical, thermal, electrical and chemical interrogation and modulation of the nervous system[8–15], conventional navigation techniques are incapable of transporting miniaturized tethered devices deep into the microvasculature primarily due to mechanical limitations. The invention of a methodology that provides rapid and safe passage for microdevices regardless of the complexity of the trajectory may become instrumental for translational medicine and neuroscience research.

Advancement of conventional catheters relies on manual adjustment of the curvature of their distal tip. To automate the navigation process, robotics research has introduced continuum devices with active control over body deformation through cable-driven mechanisms, concentric tube systems or devices that possess tuneable mechanical properties such as shape-memory and variable stiffness[16–22]. Seminal work has demonstrated the feasibility of using external magnetic fields to remotely control the pose of elastic magnetic rods[23–33]. However, navigating gently through increasingly intricate vascular networks, while avoiding lesions, requires the development of microscopic probes (μ-probes) as small as blood cells with extraordinary flexibility. This is particularly important for neurological interventions where accessing distal and tortuous vessels prohibitively increase the operation time and risk of intraoperative tissue damage. The bending stiffness of a rectangular beam scales cubically with the thickness of the material and, thus, reducing the dimensions of even high-modulus slender structures dramatically increases their flexibility. Furthermore, friction forces progressively increase with the distance, making it harder to push structures forward inside small vessels. As a result, state-of-the-art navigation techniques are not suitable for the advancement and steering of miniaturized endovascular devices.

Here, we introduce a robotic navigation strategy that relies solely on the ability of the blood flow to transport devices in vessels with arbitrary tortuosity. The method is based on the exploitation of elastohydrodynamic coupling between flexible structures and the surrounding fluid. Harnessing viscous stresses and pressure for propulsion while controlling the heading with spatially homogenous magnetic fields nullifies the need for proximal pushing. This strategy allows autonomous flow-driven transportation of μ-probes along three-dimensional (3D) trajectories close to the speed of flow (Fig. 1a). The torque exerted on the magnetic head bends the distal end of the device, which in turn interacts with the flow in such a way that the structure is conveyed to the chosen trajectory. The torque required to effectively steer the μ-probe scales favourable with vessel size and can be provided by a magnetic head as small as 40 μm under magnetic field strength, $B$, as low as 5 mT. The persistent presence of flow ensures safe advancement of the μ-probe until the next bifurcation without external intervention and with minimal contact with the walls, promoting autonomous passage through unknown or highly structured channels. In addition, in order to significantly reduce the size of the electromechanical μ-probes, we abandoned the conventional two-step navigation paradigm that is based on the use of a flexible guidewire for the advancement of the functional catheter. Instead, we designed and engineered endovascular microrobotic devices with cross-sectional area as small as $25 \times 4 \, \mu m^2$ that travel effortlessly in the vessels as if they were untethered while taking full advantage of the electrical and fluidic tethers. The physical principles of the navigation strategy have been systematically explored inside microfluidic devices with relevance to clinical challenges and using an experimentally validated computational model, which together led to the development of a repertoire of design and wireless control strategies. Finally, we demonstrate the feasibility of operating (multiple) microengineered μ-probes at the target locations for making physical measurements in custom-made biomimetic phantoms and ex vivo local injection of chemicals inside the vasculature of a rabbit ear.

## Results

**Design and operation of a μ-probe-insertion device**. Unlike conventional endovascular catheters[34], ultra-flexible and ultra-lightweight filaments cannot simply be pushed into a stream because axial compressive loads will no longer be effectively transmitted due to bending of the μ-probe. For this purpose, we developed a hydromechanical insertion system that seamlessly couples with the vasculature and keeps the μ-probe under tension during the entire operation. The proximal extremity of the μ-probe was attached to a rigid rod that slid across a sealing gasket at the rear end of a custom-made syringe barrel, and the motion of the rod was controlled by a linear positioner. Gentle pulling of the rod ensured proper loading of the μ-probe into the barrel. Saline solution was pumped inside the system through an intake, applying tensile stress to the relaxed filament. Percutaneous cannulation in conjunction with controlled perfusion was performed for the fast and versatile deployment of the μ-probe (Fig. 1b, Supplementary Fig. 1). The axial movement of the rod determined the portion of the μ-probe that was released into the target vessel. The hydrodynamic pulling forces ensured smooth passage of μ-probes into the cannula in the absence of pushing forces. Upon introduction of the μ-probe into the vessel, viscous stresses applied by the physiological blood stream dominate the propulsion and the perfusion may be terminated.

**Fluid-structure interactions in ultra-flexible μ-probes**. We first studied the deformation and passive deployment of the μ-probe in a nonuniform viscous flow with curved streamlines. Flexible electronic devices fabricated by depositing conductive elements on thin polymer substrates offer a versatile route for the development of smart μ-probes. We manufactured 4-μm-thick polyimide (PI) ribbons with 200-μm width as generic structures for the study of fluid-body interactions under the influence of flow. Coating the complete surface of the ribbon with a 100-nm-thick gold film exaggerates the stiffness of functional μ-probes with patterned electrical circuits. As a result, we guarantee that the navigation results are representative for all electronic μ-probes presented in this work. Cylindrical magnetic heads with varying sizes, diameter from 40 to 350 μm, were fabricated from a hard-magnetic elastomer composite using a moulding process. The experiments were performed inside biomimetic vascular networks made from photopolymer or elastomer using 3D printing and sacrificial moulding, respectively. A Newtonian fluid matching the viscosity of blood was pumped into the channels. We regulated the average fluid velocity, $\bar{u}$, according to the size of the vessels to stay in the physiologically relevant regime.

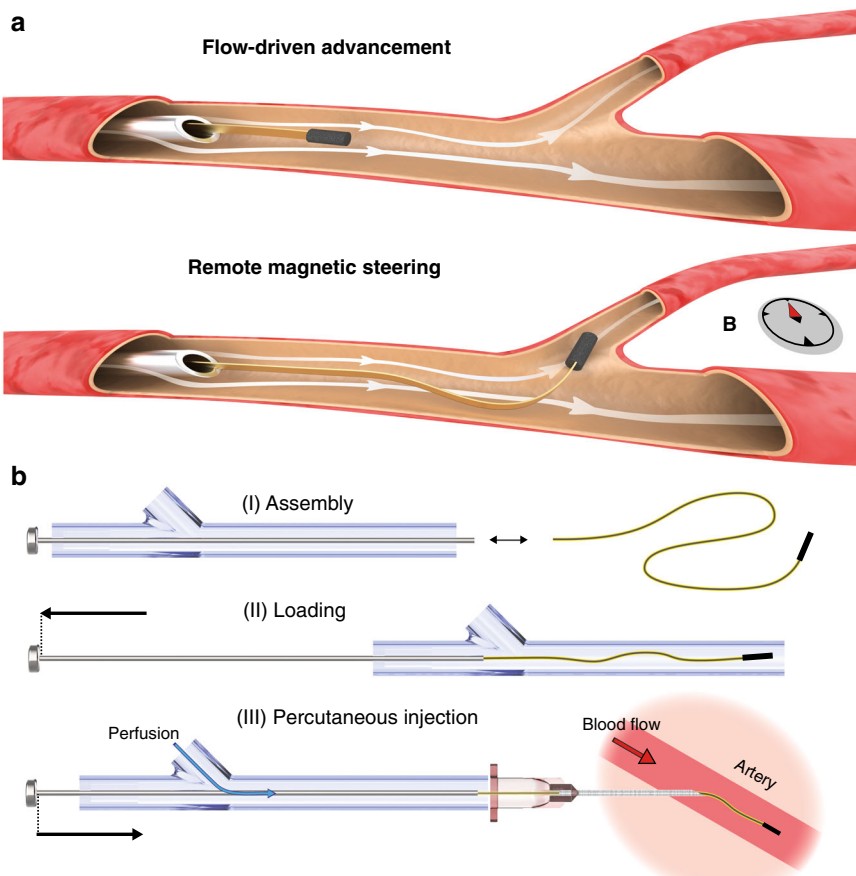

**Fig. 1 Flow-driven deployment and advancement of endovascular μ-probes. a** Ultra-flexible μ-probes are transported inside vessels by the hydrodynamic forces (top). Controlled rotation of the magnetic head using external magnetic fields steers the device into the target vessel (bottom). The small size of the μ-probe allows direct release into the vessels with standard hypodermic needles. **b** Effective deployment of the μ-probes using a proper cannulation system that overcomes mechanical instabilities. Perfusion with a physiological solution during delivery keeps the structure under tension until the flow in the artery takes over. Drawings are not to scale.

We realized a tortuous fluidic scenario to replicate challenging vascular trajectories as shown in Fig. 2a. As soon as the μ-probe was fed into the channel using the insertion device, fluid flow started to pull on the structure and kept it under tension at all times. The μ-probe was fed by moving the plunger forward and the filament perfectly followed the central streamline in the linear portion of the channel. The viscous stresses bent and transported the μ-probe through each high-curvature turn to the target location. For the chosen geometric and flow parameters, curvature up to $0.8\,\mathrm{mm}^{-1}$ and $\bar{u} = 20\,\mathrm{cm}\cdot\mathrm{s}^{-1}$, the maximal attainable velocity of the μ-probe (i.e., the maximum velocity at which we can feed the μ-probe without generating mechanical instabilities) was $2.8\,\mathrm{cm}\cdot\mathrm{s}^{-1}$, which corresponded to $0.15\,\bar{u}$ (Supplementary Movie 1). Intuitively, faster advancement can be achieved at higher fluid velocity due to the increase in viscous stresses on the μ-probe. Likewise, straighter channels allow faster deployment, pushing the maximum attainable deployment speed closer to the flow velocity. To demonstrate the instrumental role of the viscous stresses in the advancement process, we temporarily turned off the pump and kept pushing the μ-probe. The structure immediately lost tension along with hydrodynamic lift upon removal of the flow. With further pushing, the contact between the walls of the channel and the magnetic head generated mechanical instabilities such as buckling (Fig. 2b, Supplementary Movie 2). The structure unfolded and regained its straight configuration as soon as the pump was turned on, regardless of the deformed shape of the μ-probe.

The sustained tension on the μ-probe provided by the viscous stresses differentiate the proposed deployment system from conventional strategies. Standard push-based endovascular devices rely on the outward wall contact to be able to transmit the axial force along the structure in non-linear trajectories (Fig. 2c, left). The result is an important increase in friction and normal forces on the walls that would ultimately overcome the proximal insertion force and lead to advancement arrest or potentially to blood vessel wall perforation. Oppositely, in flow-driven navigation, the contact points are limited to the inward wall of the curve due to the experienced tensile regime (Fig. 2c, right). The discrete contact regions, in combination with the distributed drag force, allows maintaining a ratio larger than one between the total drag force and friction force along the entire trajectory and therefore ensuring uninterrupted advancement (Fig. 2d). The tip of the μ-probe followed almost identical trajectories while it was being pulled forward by the flow or pulled back by the insertion device (Supplementary Fig. 2). One important consideration for the retraction process is the friction at corners with high curvature. The structure must be ultra-flexible yet inextensible for effective and safe retraction through tortuous paths. Since the position of the μ-probe depends on the flow along its length, the μ-probe is not always aligned with the local streamlines. For example, when the flow has curved streamlines, the filament does not align with the flow, but rather it crosses the streamlines. A similar behaviour was reported on biological filaments forming in curved microfluidic channels[35].

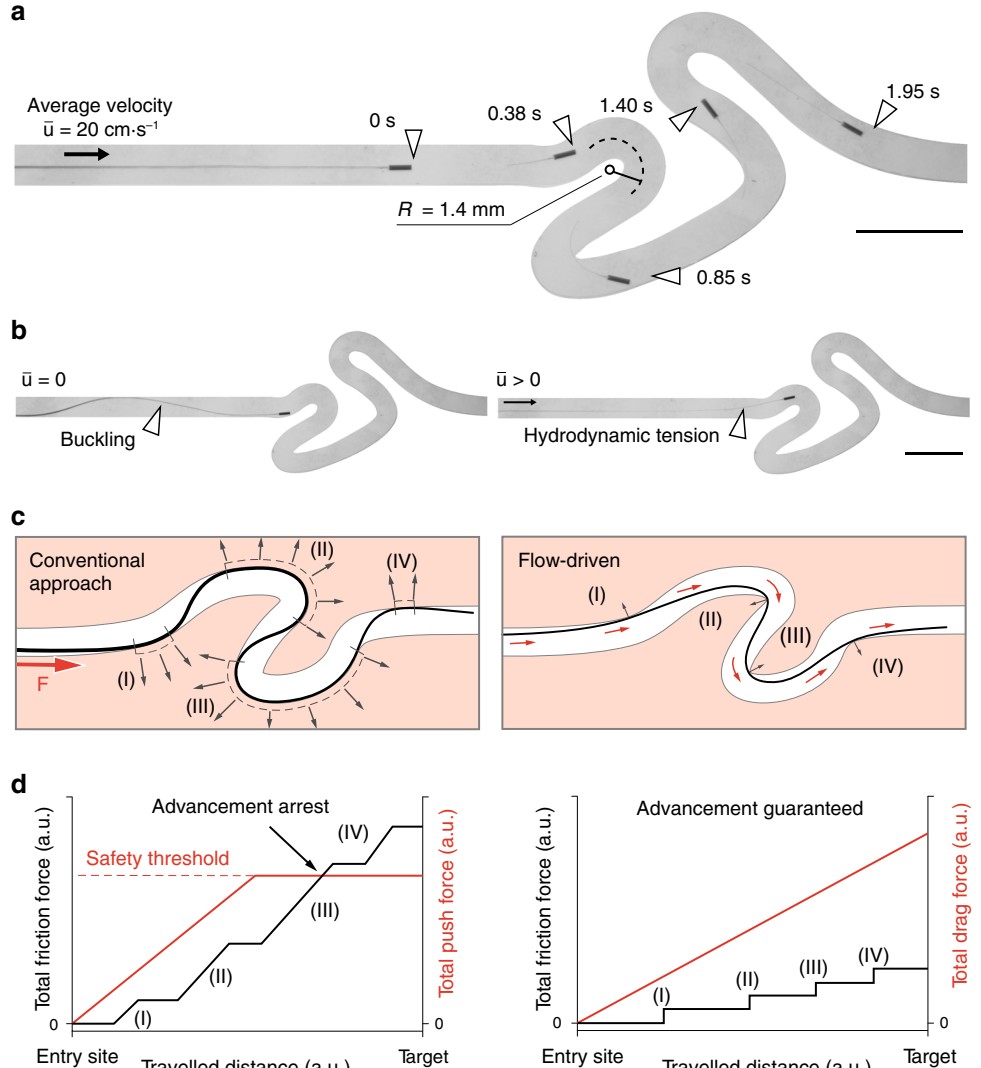

**Fig. 2 Characterization of the flow-driven autonomous advancement of μ-probes. a** Slender μ-probes are transported by the flow inside tortuous channels with high curvature at a forward average velocity of 2.8 cm·s⁻¹. The average flow velocity is given by $\bar{u}$ and $R$ denotes the radius of curvature. $R$ in the second and third turns are 1.2 and 1.8 mm, respectively. **b** In the absence of flow, pushing the μ-probe forward generates mechanical instabilities and the advancement ceases (left). Upon introduction of the flow, the μ-probe re-engages with it and continues to advance inside the channel. **c** A comparison between the conventional catheter advancement approach and fluid-driven transport introduced in this work. The conventional approach relies on a finite proximal force, $F$, and the contact pressure exerted on the outward wall (grey arrows) to transmit forces distally (left). Our approach relies instead on the tensile forces applied by the viscous stresses along the entire structure (right). Small red arrows represent drag forces. Drawings and forces not to scale. **d** In conventional navigation paradigm, friction forces increase with the length of the filament. The externally applied push force must be increased accordingly to be able to maintain the advancement. However, input forces above a safety threshold may result in vessel perforation or dissection. Ultimately, when the friction exceeds the push force, the intervention reaches its limit. On the other hand, in our approach, the continuous presence of hydrodynamic forces maintains tension on the structure and warrants a positive difference between the drag force and the friction, regardless of the travelled distance. Plots are conceptual. Scale bars, 5 mm.

We developed a computational model to gain more insight on the navigation mechanism and for rapid testing of robotic control strategies (see Supplementary Note 1 for details). The stationary pose of the μ-probe in the channel was obtained by solving a lumped model of the discretized structure while probing the fluid velocities at each node of the structure at each iteration. The simulations were based on the same channel geometry used for the navigation experiment shown in Fig. 3a, along with the measured value of the flow rate ($\bar{u} = 20\,\mathrm{cm·s^{-1}}$) and Young's modulus of the filament ($E = 3\,\mathrm{GPa}$). To reproduce the advancement of the filament inside the channel, we iteratively added material at the tip and solved the equations for the full structure. This strategy resulted in a much shorter simulation time

compared to translating the whole μ-probe as it requires significantly fewer iterations for convergence. Furthermore, this formulation was less prone to mechanical instabilities. A penalty algorithm was implemented in order to account for contact between the walls of the channel and the μ-probe. A force normal to the channel that depended quadratically on the penetration depth was applied on each node of the filament that violated the non-penetration condition. Snapshots of numerical simulations are given in Fig. 3b and an animation of the overall motion is shown in Supplementary Movie 3. While the contact points with the channel walls were predicted correctly, the simulated shape of the μ-probe showed deviations due to the simplifications of the model (Supplementary Fig. 3).

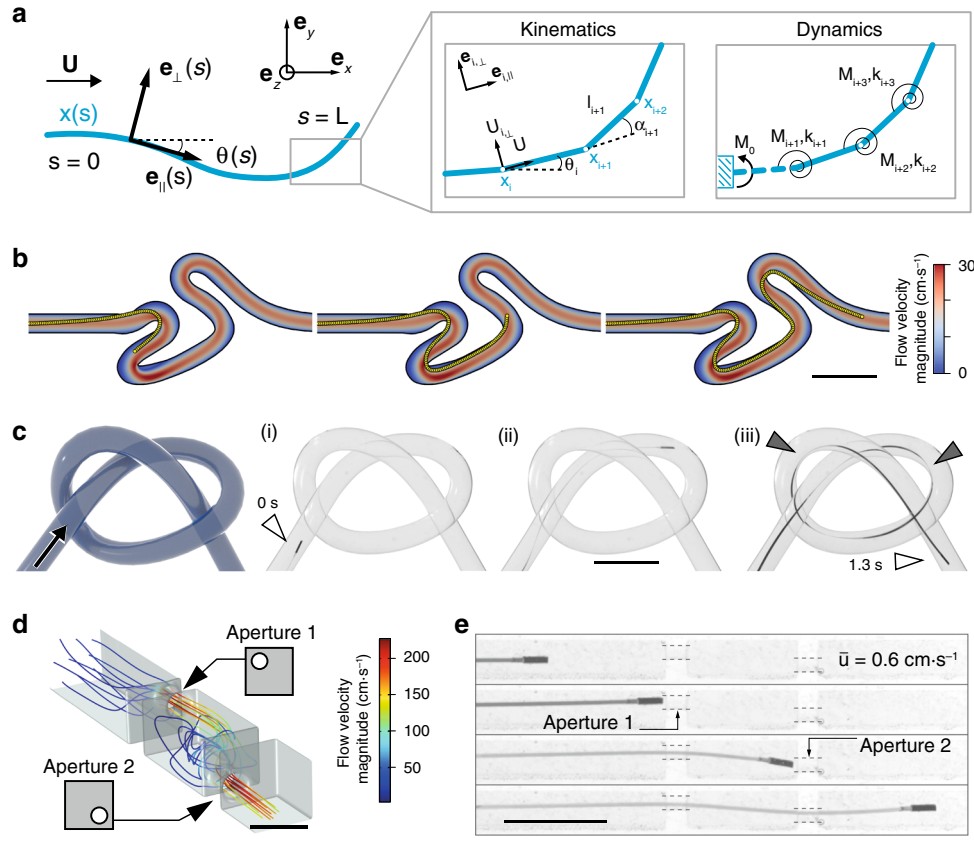

**Fig. 3 Simulation of the μ-probe dynamics and adaptive navigation. a** The μ-probe deforming in two-dimensional space is described by the centreline coordinate $s$ and a material reference frame characterized by $\{\mathbf{e}_\perp, \mathbf{e}_\parallel\}$. The fluid velocity is denoted by **U**. Discretized model of the μ-probe is shown on the right. This model is used to calculate the velocities and forces. Deformation is calculated iteratively at each node from the resultant torque, **M** and spring coefficient, $k$. **b** Simulations showing the advancement of the μ-probe in the same tortuous channel shown in Fig. 2a and Supplementary Movie 1. **c** Schematic illustration of a 3D channel with the knot shape and time-lapse images of the μ-probe (i–iii) advancing through a channel with the same geometry. The μ-probe successfully reached the target location in 1.3 s with a forward velocity of 4 cm·s⁻¹. Grey arrows indicate points of contact with the wall. **d** CFD simulation of the flow in a channel with extreme occlusions. The main channel has 2 mm × 2 mm cross-sectional area and 750-μm-diameter holes were placed on two walls in the top-left and bottom-right corner, respectively. Scale bar, 2 mm **e** Snapshots from the advancement of the μ-probe inside the structured channel simulated in (**d**), finding its way through the holes even at very low flow velocity $(\bar{u} = 0.6\,\mathrm{cm\cdot s^{-1}})$. Scale bars in (**b**, **c** and **e**), 5 mm.

**Adaptive transport of μ-probes in complex vessel phantoms.**
Using elastohydrodynamic coupling for propulsion enables enticing opportunities for μ-probes navigating inside highly curved, structured, or dynamically changing environments. We explored whether the elastic body could continuously morph in accordance with the geometry of the channels by testing the devices in a phantom with a braided channel forming a 3D knot with curvature as high as 0.25 mm⁻¹ (Fig. 3c). The μ-probe completed the track in 1.3 s at a deployment velocity of 4 cm·s⁻¹ under the control of the fluid flow with $\bar{u} = 30\,\mathrm{cm\cdot s^{-1}}$. Grey arrows indicate contact points between the channel wall and the μ-probe.

The rapid pursuit of streamlines without any external manipulation set the ground to test whether the μ-probes can pass through occlusions mimicking extreme vascular stenoses. We fabricated a channel with two internal occluding walls where the only passage was provided through two circular windows with a diameter twice as large as the size of the magnetic head. Computational fluid dynamics (CFD) simulations visualized the 3D streamlines inside the channel and quantified the distribution of flow velocity (Fig. 3d). The μ-probe autonomously navigated through both apertures while the kinematics of the motion followed the flow field imposed by the occlusions (Fig. 3e, Supplementary Movie 4). Shakes on the μ-probe head were visible in between the walls for $\bar{u}$ larger than 0.6 cm·s⁻¹ as a manifestation

of unsteady flow created by abrupt changes in channel geometry. The μ-probe moved as fast as 2.8 cm·s⁻¹ at $\bar{u} = 20\,\mathrm{cm\cdot s^{-1}}$.

**Magnetic actuation of the tip for controlled bending of the μ-probe.** While the translational motion of the μ-probe is controlled by the drag forces, the selection of a specific path requires the application of external torque. We observed that the transversal tip position of the μ-probe, $y_{\mathrm{tip}}$, right before entering a bifurcation is a reliable indicator for the direction the μ-probe is going to take (Fig. 4a). We hypothesized that by rotating the head of the μ-probe and positioning the tip at the correct location with respect to the stagnation streamline (white dashed line), we could harness the current in this new position (brown streamlines) to have the μ-probe pulled into the downstream vessel. The μ-probe would then be transported to the next bifurcation autonomously in the absence of external manipulation and the same protocol would be applied to select a new trajectory. We decided to use magnetic actuation to apply torque due to the ease of generating uniform magnetic fields, scalability arguments, and medical compatibility.

We systematically studied the effects of the magneto-elasto-hydrodynamic coupling on $y_{\mathrm{tip}}$ to develop a robust strategy for navigating μ-probes. We fabricated a representative μ-probe with a cylindrical magnetic head ($\phi_H = 350\,\mu\mathrm{m}$ and $L_H = 1\,\mathrm{mm}$) and

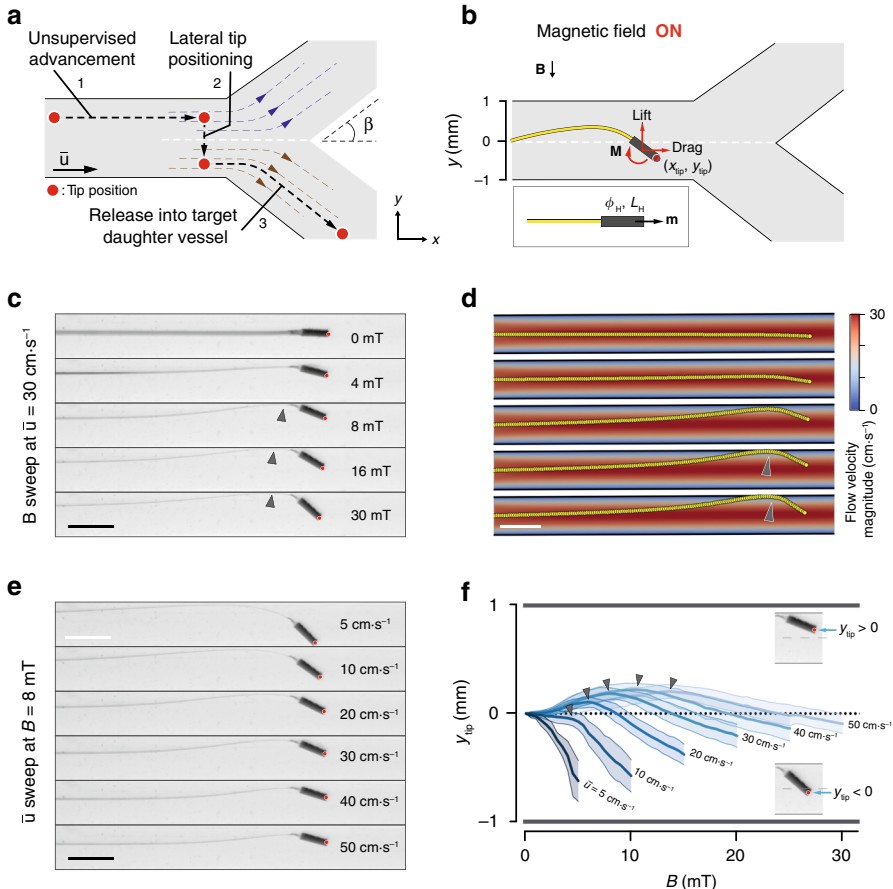

**Fig. 4 Magnetic tip steering for controlling the µ-probe position. a** Conceptual description of our proposed strategy to navigate the µ-probe into the target daughter vessel. Upon reaching a bifurcation, the µ-probe tip is laterally moved from the current stream (blue dashed lines) into the target stream (brown dashed lines) with the application of magnetic torque. The µ-probe is pulled into the stream with further release of the structure. Black dashed line shows the trajectory of the tip passing through the stagnation streamline (white dashed line) and $\beta$ denotes the bifurcation angle. **b** The transversal position of the tip, $y_{tip}$, is controlled by the magnetic actuation of the head that has length $L_H$, diameter $\phi_H$, and magnetisation **m**. Upon application of the magnetic field, **B**, the torque, **M**, exerted on the head rotates the head and bends the filament leading to an increase in both lift and drag forces. Red dot indicates the tip position. **c** The equilibrium shape of the µ-probe with increasing magnetic field at constant flow velocity ($\bar{u} = 30$ cm·s$^{-1}$). Red dot indicates the tip position. **d** Simulations of the µ-probe deformation under varying magnetic torque and constant flow velocity match the empirical results shown in (**c**). **e** The equilibrium shape of the µ-probe with increasing flow velocity at constant magnetic field ($B = 8$ mT). Red dot indicates the tip position. **f** The evolution of $y_{tip}$ in response to the fluid velocity and magnetic field sweeps. Darker lines represent the average and lighter areas represent the average ± standard deviation. Each measurement was repeated three times for both magnetic field directions and for three different µ-probes ($n = 18$). Grey arrows indicate contact with the channel wall. Scale bars, 2 mm.

used this prototype for the following characterization experiments, unless stated otherwise. Upon application of the magnetic field, the head rotated in order to align with the direction of the field, which changed the distribution of hydrodynamic forces and elastic stresses, leading to a new equilibrium configuration (Fig. 4b, Supplementary Movie 5). In the first set of trials, the µ-probe was placed in the middle of a channel at $\bar{u} = 30$ cm·s$^{-1}$ and subjected to varying magnetic fields (Fig. 4c). At low $B$, the lift force acting on the structure moved the entire µ-probe to the upper half of the channel in the opposite direction of the field. This steering mechanism closely mimics the motion of wakeboards and reaction ferries. At $B$ larger than 8 mT, the µ-probe started to contact the upper wall (grey arrow), which constrained the lift-induced upwards motion of the structure and served as a support point to further bend the distal end towards the direction of the magnetic field. At higher $B$, magnetic head rotated further, which resulted in the entry of the µ-probe tip into the bottom half of the channel. We recapitulated the µ-probe deformation under hydrodynamic forces and magnetic torque using numerical

simulations (Fig. 4d). The effect of magnetic actuation was incorporated into the computational model by discretizing the flexible head and applying magnetic moment at the corresponding nodes. We tuned the magnetization value to match the equilibrium shape at a certain $B$ and $\bar{u}$, and used the same calibrated value for the following simulations.

Next, we performed a velocity sweep at $B = 8$ mT. With increasing $\bar{u}$, the contribution of the hydrodynamic forces to the final shape also increased, resulting in a straightening of the µ-probe. As a result, the tip eventually moved from the bottom to the upper half of the channel (Fig. 4e). The overlay of the equilibrium µ-probe shapes under varying $\bar{u}$ and $B$ summarized the competing effects of hydrodynamic forces and magnetic torque on $y_{tip}$ and the steady-state shape of the µ-probe (Supplementary Fig. 4). To better visualize the effects of $B$ and $\bar{u}$ on the µ-probe shape, we plotted $y_{tip}$ for different conditions and observed a change in heading for $\bar{u}$ larger than 10 cm·s$^{-1}$ (Fig. 4f). This effect was the result of the µ-probe touching the upper channel wall, which blocked the lift-induced upward motion. Grey arrows denote the moment at which the

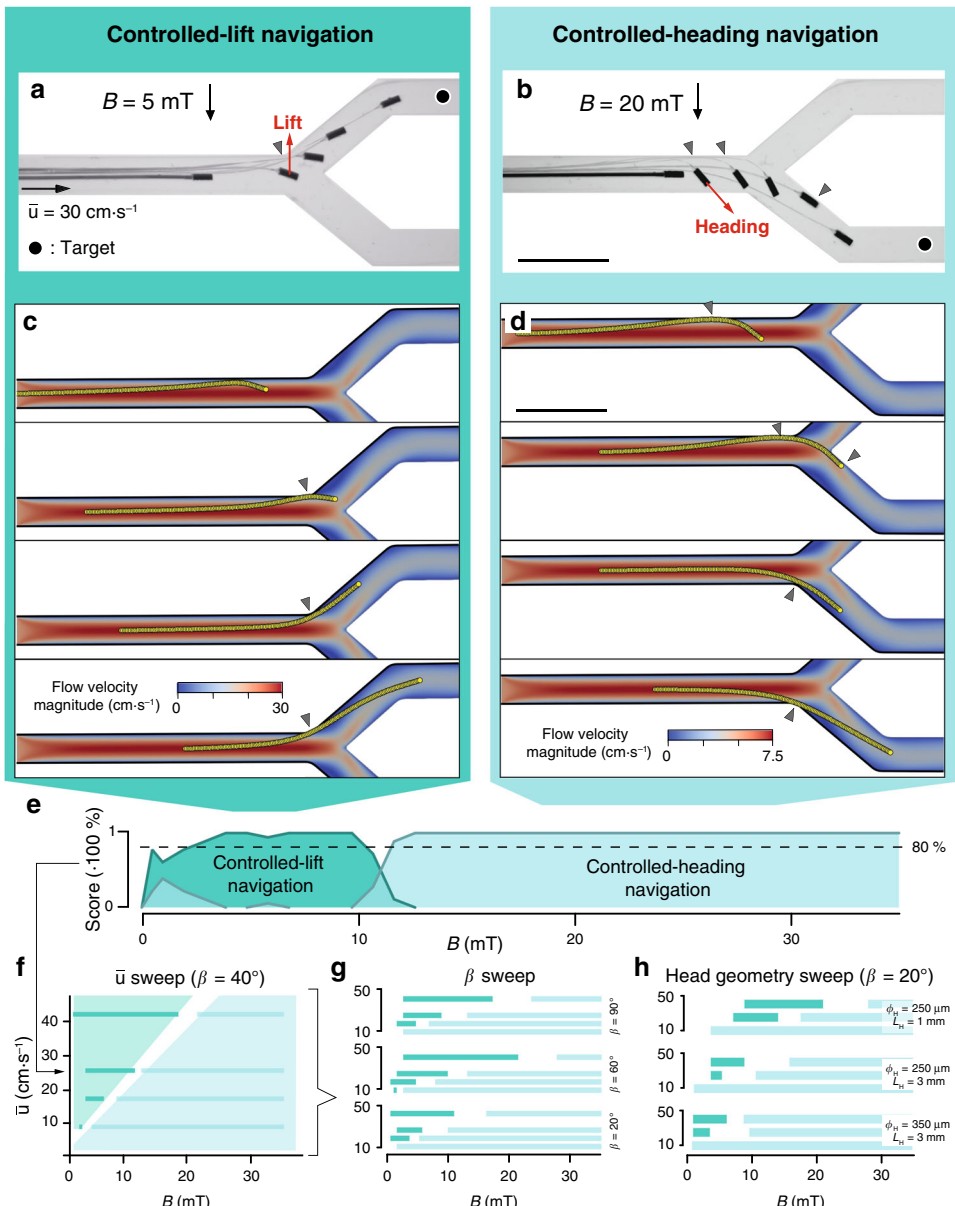

**Fig. 5 Navigation strategies to deploy μ-probes at target locations in the vasculature. a, b** Overlay of time-lapse images showing controlled transport of μ-probes to the target location (black dot) using (**a**) CL and (**b**) CH navigation. At lower $B$, CL navigation exploits the lift force emerging from the elastohydrodynamic coupling between the flow and the flexible structure. Lift force drifts the entire device into the upper daughter vessel (left). At higher $B$, the stronger magnetic torque enables setting of a course along which the μ-probe is transported by the flow (right). **c, d** Simulations showing the successful navigation of the structure adopting (**c**) CL or (**d**) CH methods. **e** A score plot quantifying the statistics of the success rate for each navigation strategy at different $B$ and constant flow velocity ($\bar{u} = 26\,\mathrm{cm \cdot s^{-1}}$). The μ-probe was placed in the middle of a 2 mm × 2 mm channel bifurcating at an angle $\beta = 40°$. Navigation tests were repeated three times for both daughter arteries with three different μ-probes ($n = 18$). **f** Phase diagram illustrating the most favourable navigation strategy for given $\bar{u}$ and $B$. The values are binarized at an 80% confidence. **g** Effects of $\beta$ on the success of each navigation strategy. **h** The influence of the magnetic head geometry on the navigation score. Scale bars, 5 mm.

μ-probe contacted the wall. The characterization experiments were repeated with magnetic heads of different sizes and geometries. Intuitively, longer and/or thicker heads could harness more lift force and require lower magnetic torque for actuation. However, longer magnetic heads did not fully exploit lift force because of the boundary conditions imposed by the geometry of the vessel (Supplementary Fig. 5).

**Controlled navigation of μ-probes in fluidic networks.** After establishing a method for effective tip positioning under flow, we tested the success of steering in the presence of bifurcations. The

μ-probe was fed into a channel that splits into two symmetrical daughter vessels with $\beta = 40°$ and $\bar{u} = 30\,\mathrm{cm \cdot s^{-1}}$. We discovered two distinct navigation strategies for selecting daughter vessels. In controlled-lift (CL), the rotation of the head drifts the structure towards the selected streamline by hydrodynamic lift (Fig. 5a, Supplementary Movie 6). Navigation based on CL is particularly appealing for situations where contact with the walls must be minimized, magnetic torque is limited, or fluid velocity is relatively high. On the other hand, in controlled-heading (CH), magnetic torque imposes the head pose while working against hydrodynamic forces (Fig. 5b, Supplementary Movie 7). CH was generally accompanied with the μ-probe contacting the channel wall, even

though the success of the navigation did not rely on this hinge mechanism. While we set the velocity of the linear positioner to 4 mm s$^{-1}$, the navigation strategies worked at velocities as high as 0.2 $\bar{u}$. We recapitulated the transport of the μ-probe with the computational model by iteratively moving the whole structure forward along the longitudinal axis. The simulation results matched major empirical observations—CL was achieved by exploiting lift to reach the target streamline and enter the daughter vessel at relatively low $B$ (Fig. 5c). CH was successfully executed at higher $B$ while the simulated μ-probe contacted the walls at similar locations (Fig. 5d, Supplementary Movie 8).

After recording the results of several navigation trials at different magnetic fields, we were able to discriminate two separate regimes favouring navigation based on CL and CH. Figure 5e shows a representative plot of the dominant navigation strategies for varying $B$ at $\bar{u} = 26\ \mathrm{cm \cdot s^{-1}}$ and $\beta = 40°$. We created a phase diagram by repeating this characterization process at different flow rates (Fig. 5f). In the phase diagram, every line corresponds to a plot that is in the form of Fig. 5e, binarized with an 80% confidence. Next, we generated concatenated diagrams by stacking phase plots for different $\beta$ (Fig. 5g) and head geometry (Fig. 5h). All results showed that the CL regime is predominant with increasing $\bar{u}$, while CH navigation had a better success rate at bifurcating angles above $\beta = 40°$ for a given velocity. Increasing the aspect ratio of the magnetic head in μ-probe design promoted CH over CL due to the restricted space for the motion of the head. Decreasing the cross-sectional area of the channel from $2 \times 2\ \mathrm{mm^2}$ to $1 \times 1\ \mathrm{mm^2}$ completely precluded navigating the μ-probe using the CL method with the chosen design and mechanical properties. Decreasing the magnetic head size and/or the bending stiffness of the μ-probe will allow navigation using the CL method in sub-mm vessels.

So far, we only showed navigation inside channels residing on a plane for detailed characterization of the deformation and motion of the μ-probe. The magnetic steering was easily extended to 3D for a complex channel geometry that was provided to the user prior to the operation (Supplementary Fig. 6). Alternatively, teleoperated 3D navigation was performed using the visual feedback provided by two cameras observing the workspace from orthogonal views (Supplementary Fig. 7). Controlled application of magnetic torque only when the μ-probe approaches bifurcations would suffice for reaching target locations. On the other hand, our method may fail when the flow in the chosen branch has significantly lower velocity compared to the alternative route. Fortunately, magnetic continuum devices provide a rich repertoire of actuation modes that can generate propulsion inside stationary fluids[36–38]. By actuating the head with time-varying magnetic fields, we facilitated entry into hydrodynamically unfavourable vessels (Supplementary Note 2). Notably, we can navigate μ-probes in tight vessels that are slightly larger than the magnetic head. To demonstrate this feature, we released μ-probes with 250 μm head into a 300 μm channel. The structure barely bent under magnetic actuation due to the confined space yet the μ-probe successfully navigated through bifurcations (Supplementary Fig. 8, Supplementary Movie 9). With further miniaturization, navigation inside microfluidic channels as small as 100 μm in width became feasible. We successfully transported μ-probes with a size of 25 μm × 4 μm and a 40-μm diameter magnetic head in <2 s (Supplementary Fig. 9). Using μ-probes within a similar size range, we may reach distal microvasculature in the body, a feature that is currently impossible to achieve. Finally, our navigation strategy is compatible with the deployment of multiple μ-probes. We iteratively steered four μ-probes to pre-determined target locations in a phantom with three bifurcations (Supplementary Fig. 10). The profile of the flow around the existing μ-probes autonomously creates alternative paths for the incoming μ-probes. Thus, the number of μ-probes that can be simultaneously deployed is only limited by the relative size of the μ-probe with respect to the vessel diameter.

**Local characterization of flow using electronic μ-probes.** After establishing the navigation strategy, we explored whether we could dynamically record physiological parameters such as electric potential, temperature, or flow characteristics using μ-probes. PI substrate was used to fabricate the main chassis while titanium and platinum strips were deposited to form electrodes and electrical circuits, respectively. As a proof of concept, we engineered a μ-probe that uses convective heat transfer to measure the characteristics of local fluid flow. Upon injection of current into the heater circuit, the generated heat is transferred to the sensor circuit at a rate determined by the velocity profile of the fluid. The fabricated electronic μ-probes consisted of a 0.1 mm × 2 mm heating element and a 200 μm × 500 μm sensor component, positioned 50 μm apart from each other (Fig. 6a). Conventional flow sensors that are based on heat transfer operate either in continuous mode or pulsed mode[39–42]. In continuous mode, the change in the sensor resistance with the flow is monitored. As a result, this measurement requires high signal-to-noise ratio (SNR), making it challenging to be implemented on miniaturized devices. Our flow sensors were operated in pulsed mode, by measuring the time-of-flight (TOF) between the injection of current to the heater and the detection of peak resistance at the sensor. This method requires a high sampling rate, which does not pose a trade-off with miniaturization.

The input to the heater was a 10-ms current pulse with 4 mA peak value, and the sensor resistance was recorded at 5 kHz sampling rate (Fig. 6b). A single temporal readout was acquired within 150 ms under physiologically relevant fluidic regimes. The speed of sensing allowed real-time interrogation of the flow during navigation. We performed a series of measurements at the wall of a channel with $2 \times 2\ \mathrm{mm^2}$ cross-sectional area at varying fluid velocities. The data showed an exponential decrease in the TOF with increasing $\bar{u}$ (Fig. 6c). However, the extracted calibration curve gave inaccurate results in channels with different cross-sectional area. The measurements were strongly influenced by the position of the μ-probe as a manifestation of the parabolic flow profile. Notably, the small ratio between the thermal over the flow boundary layer thickness dictates that the convection of generated heat is governed by the velocity gradient on the surface of the μ-probe. As a result, we calibrated the TOF with respect to the wall shear stress, $\tau_{wall}$, which captures the effects of channel geometry on the thermal convection. The wall shear stress was computed from the fully developed flow profile, which was extracted using the Fourier sum approach[43] (see Supplementary Note 3). Finite element method (FEM) simulations quantitatively showed that the measurements were indeed highly local around the sensor unit (i.e., thermal boundary layer much smaller than flow boundary layer). Thus, the heat convection was dictated by the flow profile and hence the $\tau_{wall}$ (see Supplementary Note 4).

To validate the calibration curve at different channel geometries and flow conditions, we made on the fly measurements of $\tau_{wall}$ while being navigated within structured channels. The μ-probe was deployed into a stenotic region where the cross-sectional area of the channel was reduced from $2 \times 2\ \mathrm{mm^2}$ to $1 \times 1\ \mathrm{mm^2}$ (Fig. 6d). Empirical data recorded at the pre-stenotic and stenotic regions varied only by 2.1% and 13.9% from the analytically calculated values (Fig. 6e and Supplementary Table 1). The relative increase in $\tau_{wall}$ was approximately eightfold as expected from the relation $\tau_{wall} \propto \bar{u}/\mathrm{h}$ where $h$ is the height of the

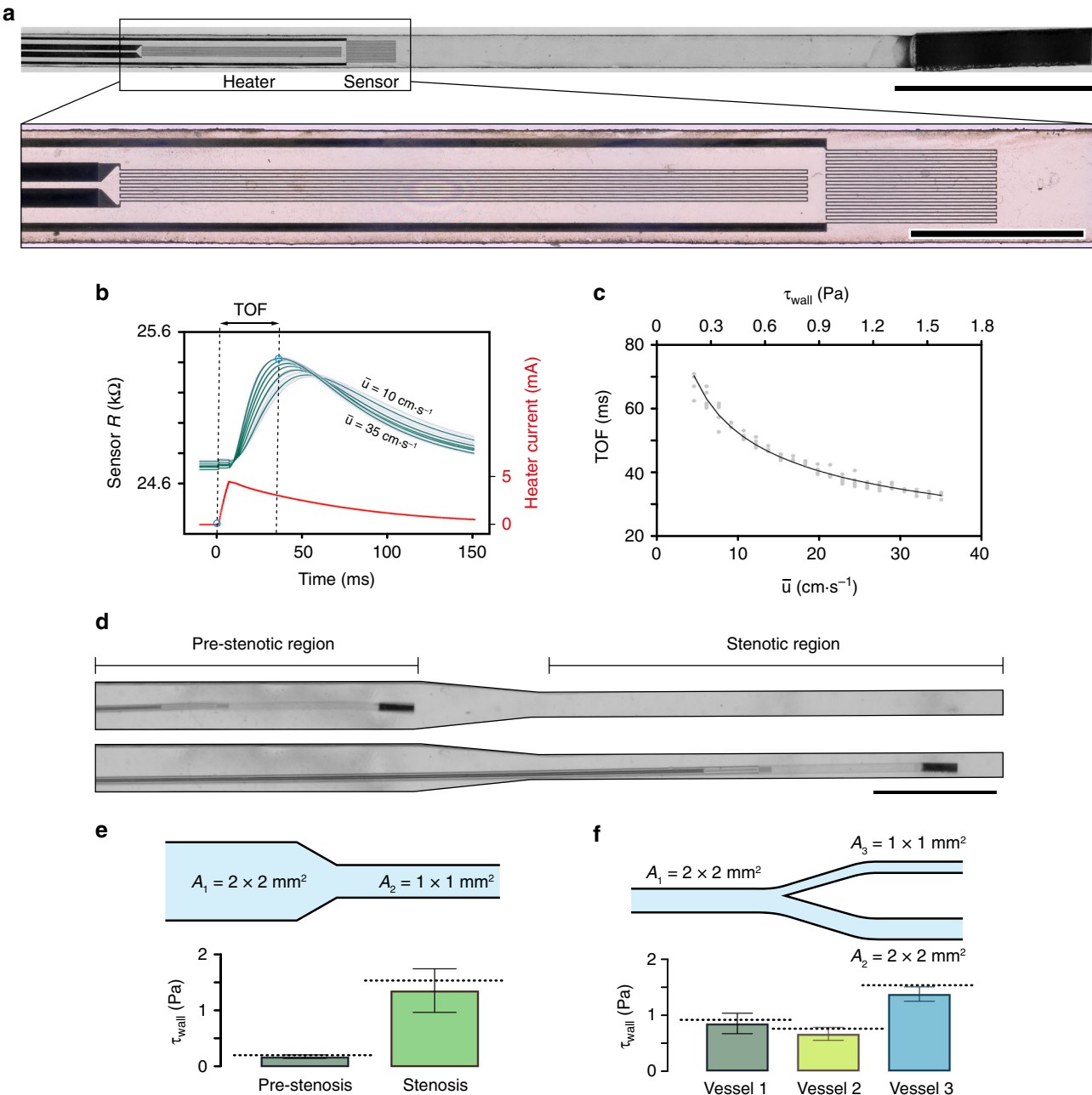

**Fig. 6 Development and navigation of microengineered thermal flow sensors. a** Picture of the flow sensing μ-probe showing the heater and sensor elements along with the magnetic head. Scale bar, 1 mm. Inset shows the details of the heater and the sensor circuits placed 50 μm apart from each other. Scale bar, 500 μm. **b** Temporal variation in the resistance of the sensor is recorded and compared with the input signal to measure the TOF of the heat pulse. Dark lines represent the average of 30 consecutive measurements that are shown in light curves. The TOF is measured for varying $\bar{u}$ in a 2 mm × 2 mm channel. **c** Calibration curve of TOF with respect to $\bar{u}$ and the $\tau_{wall}$ in a 2 mm × 2 mm channel. Black curve represents the average of 6 different tests. **d** Representative images showing the μ-probe at a pre-stenotic region and inside the stenosis. The cross-sectional area decreases 4 times inside the stenotic region. Scale bar, 5 mm **e** Wall shear stress measurements taken in the stenotic model shown in (**d**), after calibration with the curve shown in (**c**). The histograms are an average of 8 different trials with 30 measurements made at each trial. The dashed lines show the analytical calculations; error bars represent the average ± standard deviation. **f** Wall shear stress measurements taken in a bifurcating channel at three different locations (black bullets), after calibration using the curve shown in (**c**). Histograms represent 10 different trials with 30 measurements at each trial.

channel. As a final demonstration, we verified that magnetic actuation and bending of the filament did not interfere with the flow measurements. The μ-probe was navigated in a channel with two branches that had a different cross-sectional areas (Fig. 6f and Supplementary Fig. 11a). The μ-probe was successfully positioned at both branches, and the shear stress measurements were close to the calculated values with 4.6%, 8.5%, and 9.0% error, respectively (Supplementary Table 1). The repeated

measurements performed at different locations and with three different μ-probes showed reliable measurement and manufacturing (Supplementary Fig. 11b).

**Ex vivo demonstration using μ-probes and microscopic catheters (μ-catheters).** To test the feasibility of the navigation of the μ-probes under physiological conditions, we performed ex vivo tests

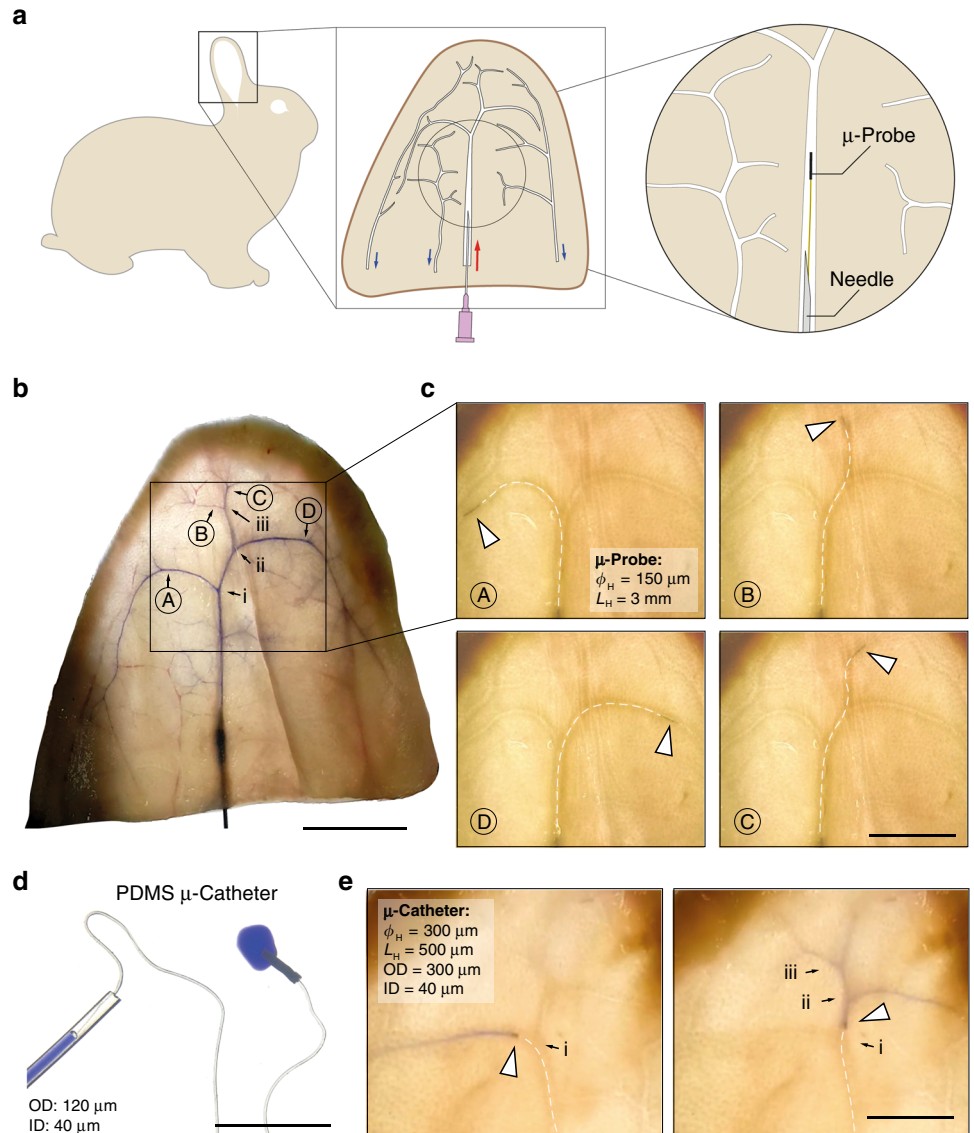

**Fig. 7 Navigation and operation of μ-probes in perfused ex vivo rabbit ears. a** Schematic representation of the vascular system in the rabbit ear. Red arrow marks the central artery ramification and blue arrows indicate the venous system. **b** Ink-perfused ear at the proximal central artery, highlighting the main arterial branches. Bifurcation is marked with roman letters (i, ii, and iii) while the targets locations are marked with capital letters (A, B, C, and D). Scale bar, 20 mm. **c** A μ-probe with 150-μm diameter and 3-mm long magnetic head was proximally inserted into the vasculature using the perfusion-based insertion system at a flow velocity between 0.5 and 1.5 cm·s$^{-1}$. White dashed lines outline the μ-probe for clarity. Scale bar, 10 mm. **d** A picture of the magnetic μ-catheter tube. A blue ink is injected through the μ-catheter. Scale bar, 5 mm. **e** Positioning of the μ-catheter head at target locations and localized injection of the ink. A total of 9 rabbit ears were used. Scale bar, 10 mm.

on perfused rabbit ears (Fig. 7a). The favourable transparency of the tissue allows standard camera imaging of μ-probes without requiring advanced medical angiography systems. Moreover, the subcutaneous position of the vessels enables the percutaneous insertion of the μ-probes with our hypodermic insertion device. We first injected a blue ink into the central artery to map the arterial system (Fig. 7b). Four target locations were chosen (denoted by capital letters) and three layers of bifurcations were marked (i, ii, and iii). The ear was perfused with a saline solution at a constant flow rate between 30–90 μL s$^{-1}$. A μ-probe with $\phi_H = 150$ μm and $L_H = 3$ mm and 4 μm × 250 μm body cross-section successfully reached all target locations through CH navigation under 3 s with an average advancement velocity of 1 cm·s$^{-1}$ (Fig. 7c, Supplementary Movie 10). So far, to navigate inside biomimetic phantoms, we applied a magnetic field that is perpendicular to the orientation of the main artery and tuned the

strength of the magnetic field to guide the tip to the chosen daughter artery. Here, we applied a magnetic field with constant field strength and dynamically tuned its orientation to align the μ-probe head along the direction of the chosen daughter artery (Supplementary Fig. 12). We anticipate that this strategy will be easier to implement in clinics. The μ-probe entered all the physically fitting vessels, with no apparent friction arising from vessel walls and without causing any perforation. In fact, ex vivo tests revealed excellent lubrication capabilities compared to in vitro experiments.

Slender filaments in different forms and material composition may further extend the capabilities of the proposed concept of flow-driven navigation. Conventional catheters are manufactured as tubes that enable injection of contrast agents and embolization agents or even allow the collection of blood clots. Motivated by these capabilities, we fabricated elastomer microtubes using a

thermal polymerization technique. A tubular magnetic head was glued to the distal end of the flexible tube to complete the μ-catheter. This protocol allows fabricating Polydimethylsiloxane (PDMS) μ-catheters with an outer diameter as small as 120 μm and several cm-long (Fig. 7d). Upon perfusion through the insertion system, the hydrodynamic forces transported the flexible μ-catheter inside the vessels of the rabbit ear, and magnetic steering allowed the device to navigate through bifurcations. To demonstrate the potential for endovascular embolization or targeted drug delivery, we temporarily stopped the perfusion and injected a blue dye at various locations within the vascular network (Fig. 7e and Supplementary Movie 11). The diameter of the vessels, where the dye injection was performed, were estimated from the evacuation speed of the dye upon re-activation of the perfused flow. Assuming equal flow distribution at the first two bifurcations and no other prior bifurcation, we estimated the diameter of the artery at the point D to be 300 μm. Experimental observation corroborated with this estimation; the μ-probe in Fig. 7c got stuck at point D, indicating a vessel diameter in the order of the largest feature of the μ-probe, which was 250 μm.

**Translation of the technology to in vivo applications**. By actuating the head of the μ-probes with uniform magnetic fields that are relatively weak and vary slowly, we provided a technology that can be directly translated to in vivo studies. Human-sized magnetic control systems such as Stereotaxis Niobe and Aeon Phocus are capable of generating magnetic fields with the required strength, complexity, and speed[24,44]. On the other hand, interventional neuroradiologists usually perform endovascular operations using the visual feedback provided by a fluoroscope. To this end, we verified the visibility (i.e., radiopacity) of the magnetic head inside an ex vivo rabbit head and an anthropomorphic head phantom using two different fluoroscopes that are regularly used in the operating room (Supplementary Fig. 13). We could track magnetic structures as small as 75 μm while more advanced systems can resolve even smaller structures. The image quality can be further enhanced with the incorporation of radiocontrast agents such as iodine and barium into the magnetic head. Thanks to the minimal external control required by our navigation strategy, it would be possible to drastically reduce X-ray exposure. Another potential concern is the magnitude of forces applied to the vessel walls during the advancement and retraction of the μ-probe. We built an analytical model by approximating the configuration as a belt friction system[45] and developed a series of experimental platforms that can report traction forces on the μ-probe during navigation inside the phantoms (Supplementary Note 5). The results showed that the normal and tangential forces are on the order of $10^{-4}$ N while the forces recorded with conventional endovascular catheters are on the order of $10^0$ N[46,47].

In both in vitro and ex vivo experiments, we were perfusing the vessels with a saline solution, and the same perfusion system was coupled to the insertion device for the deployment of the μ-probe. In in vivo trials, the injection device must engage to the existing cardiovascular pumping system through proper cannulation. We devised a deployment system for subcutaneous insertion of the μ-probes using standard flexible cannulas (Supplementary Fig. 14). Analogous to the previous version of the insertion system shown in Fig. 1b, the system comprises of a rigid rod for controlling the deployment of the μ-probe, a commercially available Y-shape adapter for perfusion, and a flexible cannula for the injection. The flow was provided by an IV-bag instead of the peristaltic pump. The hydrostatic pressure was sufficient to provide a gentle flow that drags the μ-probe into the vessel. We have also developed a

hybrid toolkit that combines our technology with commercially available catheters and guidewires. Here, the μ-probe was attached to the tip of a 300 μm guidewire that moves inside a 3F (French) catheter (Supplementary Fig. 15). Proximal perfusion through a 5 mL syringe or a pressurized IV-bag was again sufficient to provide the drag forces to keep the μ-probe in tension. The main advantage of this modification is the compatibility of the technique with standard surgical protocols. As a final remark, clot formation during navigation in blood vessels is a critical risk factor. We evaluated the anticoagulant properties of our μ-probes using fresh human blood. The μ-probes induced neither platelet activation nor platelet aggregation, revealing that the polymers that the chosen polymers do not induce thrombus formation in their original finishing (Supplementary Fig. 16). The surface properties of the μ-probes may be modified if needed using an additional bioinspired omniphobic surface coating, that was previously shown to drastically enhance anti-fouling and anti-coagulation properties of medical devices[48].

## Discussion
The ability to reach very peripheral vessels or to access those that are currently too small to be catheterized may allow endovascular specialists to penetrate the so-called perforating arteries, thereby giving access to therapeutic options in structures such as the brain stem, the retinal arteries or the basal ganglia. Such access might open new therapeutic options to treat deep-seated or very peripheral tumours inside the brain, and target thromboembolic diseases. The technology we have developed may open new perspectives in the work up and management of neurological disorders such as epileptic seizures. Moreover, the concept of transporting multiple μ-probes with the flow will allow easier penetration in the depths of arteriovenous malformations. Other delicate structures that could benefit from enhanced atraumatic peripheral navigation are the spinal arteries that are known to be fragile and dangerous to catheterize. Aside from clinical potential, we foresee a clear path on long-term recording and stimulation of neural tissues through arterial access. Seminal work has shown that these tasks can be performed in cerebral veins using large stent-shaped electrodes[6,7]. The ability to simultaneously deploy multiple leads will enable 3D mapping of activity and spatio-temporally controlled activation at multiple sites. Optogenetics emerged as a versatile technique to address numerous scientific questions in neuroscience[15,49,50], however, state-of-the-art stimulation devices are bulky and invasive. With our technology, for the first time, we showed that accessing deep brain regions through the endovascular path is technically feasible. This will enable surgery-free optogenetic stimulation using designer μ-probes.

The μ-probes were fabricated using conventional clean-room technology, thus, they can be produced in large numbers and decorated with sophisticated electronic circuits[51]. While thin-film techniques manufacture electronic μ-probes with high fidelity, the space provided by silicon wafers is currently imposing a limit in the total length of the devices. Alternative manufacturing techniques such as thermal drawing[52,53] and 3D printing[54] enable production of metre-long smart fibres and electronic devices. The navigation paradigm is not dependent on the material choice. We can tune the bending stiffness of microfabricated devices by changing their design so that flow-driven navigation would work. We demonstrated this versatility by manufacturing devices from a wide range of stiffnesses, PDMS (Young's modulus: 10 MPa) and Kapton (Young's modulus: 4 GPa). Devices can be fabricated from medical-grade polymers that are used to manufacture commercially available catheters such as polyurethane and polyethylene. The design paradigm can also be adapted to hydrogels

such as gelatin and N-isopropylacrylamide (NIPAAm), opening up the floor for biodegradable electronic devices[55] and programmable soft micromachines[56].

## Methods

**Experimental platform.** The experimental setup consists of an electromagnetic manipulation system, a linear positioner, and a fluid circuit (Supplementary Fig. 17). The 3-axis nested Helmholtz coil system was designed following the instructions provided elsewhere[57]. Homogenous magnetic fields are generated within a volumetric space of $94 \times 53 \times 22$ mm$^3$ with a maximum $B$ of 50, 60, and 85 mT, respectively. The current flowing through the coils is provided by three sets of power supplies (S8VK 480 W, Omron and SM 52-AR-60, Delta Elektronika) and servo controllers (Syren25 and Syren50, Dimension Engineering), which are modulated using an analog/digital I/O PCi Express board (Model 826, Sensoray). The off-the-shelf motorized stepper-motor linear positioner controls the forward and backward motion of the connecting rod and, as a result, the position of the μ-probes. Teleoperation of μ-probes was performed using a 3D mouse (3DConnexion) and keyboard. The fluid flow was provided by a peristaltic pump (LP-BT100-2J, Drifton). The oscillations in pressure were dampened using a Windkessel element. A solution of 42.5 wt% of glycerol (Sigma-Aldrich) in deionized water was used as a blood analogue. Visualization of the μ-probes inside the phantoms was performed using CMOS camera (acA4024-29uc, Basler). We built an illumination system from smartphone backlights (Supplementary Fig. 18).

**Fabrication of biomimetic phantoms.** Phantoms were fabricated using two different techniques. The first method allows fine control over the design of complex channels that reside on a single plane. Devices drawn with CAD software (Autodesk Fusion360) were printed using a 3D stereolithography printer (Formlabs 2) from a transparent resin (Clear resin, Formlabs) (Supplementary Fig. 19a, b). The surface of the printed piece was coated with a UV-glue (NOA81, Thorlabs) to optimize optical transparency (Supplementary Fig. 19c). A second method that is based on a recently published report[58] was developed for fabricating 3D phantoms (Supplementary Fig. 19d). A sacrificial negative mould was fabricated from 1.75-mm diameter water-soluble poly vinyl alcohol (PVA) filaments used in extrusion 3D printing. The PVA filament can be manually shaped using the thermoplastic properties of the material. The filament was heated above the glass temperature with a heat gun prior to shaping. Increased temperatures allow the bonding of filaments and the formation of bifurcations. The PVA mould was then casted inside PDMS (Sylgard 184, Dow-Corning) scaffold prepared at a 10:1 mass ratio (prepolymer: curing agent) and partially cured at room temperature overnight. A final curing step in the oven at 65 °C for 2 h ensured complete polymerization without deforming the PVA mould. The PVA mould was dissolved in water in a sonicating bath (CPX3800H-E, Bransonic) at 60 °C for a few days. Off-the-shelf pen ink was injected into the channels to enhance optical contrast between fluidic channels and the background (Supplementary Fig 19e).

**Computational model.** The filament was discretized in $n$ elements of lengths $l_i$ as shown in Fig. 3a. The fluid velocity at each node was projected onto a local coordinate system. The Stokes hypothesis does not hold for the whole range of Reynolds numbers encountered in the experiments (Re∼0.01–1000). Therefore, viscous forces at each node, $F^i_\perp$ and $F^i_\parallel$ are calculated from the normal and tangential components of the velocity at that node, $U^i_\perp$ and $U^i_\parallel$, using an extended resistive force theory[59] that takes inertial effects into account. The proximal tip of the μ-probe was clamped while the distal tip was kept free. A lumped model for the discretized filament was derived from the pure bending beam equation. Each node has a torsional stiffness $k_i = \frac{EI}{l_i}$ where $E$ is the elastic modulus and $I$ is the area moment of inertia. We determined the relative bending angle at each node from the total torque generated by elastic, magnetic, and viscous stresses. An iterative method[60] was implemented to solve the lumped model for large macroscopic deformations. We refer the reader to Supplementary Note 1 for the details of the formulation.

Two-dimensional (2D) CFD simulations of the flow inside the vessels were performed using an open-source software (OpenFOAM). The CAD design of the phantoms was used to extract the boundaries of the vessels. The cartesian2DMesh module of cfMesh was used to create the two-dimensional meshes of the channels with boundary layer elements. The initialisation of the flow was done using potentialFoam and the incompressible steady-state laminar Navier-Stokes equations were solved with simpleFoam. 3D CFD and FEM simulations were performed using COMSOL Multiphysics software.

**Fabrication of electronic μ-probes.** Devices were fabricated using standard microfabrication techniques. The details are provided in Supplementary Note 6. In brief, generic μ-probes have been prepared by spin-coating on a 4-inch Si wafer a 4 μm-thick layer of PI (PI2610 Hitachi Chemical DuPont MicroSystems GmbH). A positive photoresist (AZ1512, 2 μm exposed by Heidelberg Instruments MLA150, 405 nm and 104 mJ cm$^{-2}$ and developed by AZ 726 MIF) has been used as mask for sputtering (Alliance Concept AC450) a 100-nm-thick layer of gold in

stripes of 250-μm width and 9-cm length, followed by lift-off in acetone. The μ-probe borders have been then laser cut (Optec MM200-USP) and detachment from the wafer has been done manually. The fabrication of the flow sensors involves the sputtering of titanium and platinum layers serving for adhesion and electrical conduction, respectively. To add the two serpentines, designated as temperature sensor and heater, another 25-nm thick layer of platinum has been sputtered over the wafer, after surface activation by argon plasma (Alliance Concept AC450). The shapes of the traces and serpentines were defined using spin-coating and removal of positive photoresists (AZ9260, 8 μm and AZ1512, 2 μm). The electrical circuits were sandwiched between PI layers.

The magnetic head for all μ-probes was fabricated from a composite of PDMS and Neodymium Boron Iron (NdFeB) microparticles with an average diameter of 5 μm (Magnequench, Germany), mixed at a mass fraction of 1:1. The structures were cured at 65 °C in the oven in a custom-made mould. The size of the cylindrical magnetic heads varied from 40 to 350 μm in diameter and from 100 μm to 3 mm in length. Magnetization was performed using an impulse magnetizer (Magnet-Physik) at a field strength of 3500 kA m$^{-1}$. The magnetic head was attached to the distal end of the Kapton ribbon using epoxy and manual positioners (Thorlabs). The electrical measurements were performed using a sensitive dual-channel source measure unit (Keysight B2902A).

**Fabrication of μ-catheters.** Microfluidic μ-probes were fabricated by following a simplified version of a previously described protocol[61]. In brief, a 40-μm-diameter tungsten wire (Goodfellow) was immersed in PDMS (10:1 ratio) and heated with the electrical current at a density of 320 A mm$^{-2}$. Thermal dissipation provided by Joule's heating ensured local polymerization around the wire. The thickness of the PDMS μ-catheter was determined by the time of polymerization (Supplementary Fig. 20). The wire coated with the polymerized PDMS was then immersed in an acetone bath and vigorously agitated to remove uncured PDMS. After completing curing in an oven at 65 °C for 2 h, the tungsten wire was first gently and mechanically disengaged from the PDMS coat using tweezers. Next, to apply a distributed load, the PDMS μ-catheter was sandwiched between two sheets of PDMS and the tungsten wire was gently pulled out while applying pressure on the PDMS sheets. A tubular magnetic head was bonded to the distal end of the μ-catheter using PDMS as a glue and a tungsten wire to prevent clogging.

**Ex vivo experiments.** Navigation experiments were performed with ex vivo Rex rabbit (weight between 2.8 and 3.5 kg, age ∼4 months) ears immediately following their delivery. The ears were perfused with phosphate-buffered saline (PBS) solution 1X (Sigma Aldrich) in the central artery shortly after excision. The μ-probes were introduced into the central ear artery using the insertion device and a 21 G hypodermic needle. A $3 \times 10^{-3}$ w/v% of Pluronic-F127 solution (Sigma Aldrich) was added to the PBS solution to avoid adhesion of the PDMS tubes to the barrel of the insertion system. Mean perfusion flow was maintained at a rate of 90 μL s$^{-1}$. A total of nine ears were used throughout the experiments. The ears were obtained from a local farm (La Ferme Aebischer, 1595 Faoug, Switzerland).

**Platelet aggregation and activation.** Fresh human blood samples were collected from the Health Point at EPFL. Platelet-rich plasma (PRP, 300 μL) was added to the wells containing the devices, and the plate was agitated for 15 min, 1200 RPM at room temperature. Platelet aggregation was measured using a plate reader (595 nm – PerkinElmer, Victor X3). For the platelet activation test, the ATP release from activated platelets was measured using luciferin-luciferase reaction, as previously described[62]. In brief, Luciferin-luciferase ATP reaction buffer (ATP mix) was prepared using 200 μg mL$^{-1}$ of luciferin, 40 μg mL$^{-1}$ of luciferase, 2.4 mg mL$^{-1}$ of bovine serum albumin, and 2.4 mg mL$^{-1}$ MgSO$_4$. 100 μL ATP mix were added into the well containing the samples previously agitated for 15 min, 1200 RPM at room temperature before measuring platelet aggregation using absorbance. Four independent experiments were performed for both tests using blood from four different donors ($2.75 \times 10^5$ μL$^{-1}$ to $3.66 \times 10^5$ μL$^{-1}$ – PRP concentration among donors). Collagen-I (10 μg mL$^{-1}$) was used as positive control and PRP incubated with PBS was used as negative control. All protocols were approved by the Swiss ethics committee (project-ID 2017-00732).

**Fluoroscope images.** X-ray images of the magnetic structures were taken using two different fluoroscopes, Canon Alphenix Sky+ with 200-μm resolution and Canon Alphenix Core+ with 76-μm resolution, inside an ex vivo rabbit head or an anthropomorphic head phantom.

**Reporting summary.** Further information on research design is available in the Nature Research Reporting Summary linked to this article.

## Data availability

Data supporting the findings of this study are available within the paper and its Supplementary Information files. All other relevant data are available from authors upon reasonable request.

## Code availability

All relevant code is available upon request from the authors.

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

## Acknowledgements

This work was supported by the European Research Council (ERC) under the European Union's Horizon 2020 research and innovation program (Grant agreement No. 714609). We thank Dr. Pedro Reis for giving access to their magnetizer, Dr. John Kolinski for invaluable discussions, David Aebischer for providing the rabbit ears, Erik Mailand for his assistance with the biocompatibility assay, and Heidi Olsen from Canon Medical Systems Europe for her assistance with fluoroscope imaging. We also thank Fatemeh Farsijani, Benoît Pasquier, Sven Montandon, and Vincent de Poulpiquet for their technical support.

## Author contributions

L.P., D.G., and M.S.S conceived the original idea. L.P., P.D., and M.S.S designed the experiments. P.D. formulated and implemented the computational model, L.P. performed the experiments and analysed the data, A.F. and D.G. developed the electronic μ-probes, A.M.L., and N.S. performed the anticoagulation assay, P.J.M. performed the guidewire navigation experiments, L.P. and M.S.S. wrote the manuscript with contributions from all authors, M.S.S supervised the research.

## Competing interests

L.P., P.D., and M.S.S. filed a patent (PCT/IB2020/056374) on the ultra-flexible flow-directed device and system. The authors declare no other competing interests.
