## [Peer Review File · Nature Communications]

REVIEWER COMMENTS

Reviewer #1 (Remarks to the Author):

In this manuscript, tethered ultra-flexible endovascular micro-probes that are three orders of magnitude smaller than the smallest microcatheter are developed. These probes can be steered and guided to areas of the vascular network otherwise not accessible by conventional catheters used in the clinic today. While this is very exciting with great results, I have several concerns that could be addressed in the introduction or in the discussion:

1. In the introduction it is said that the microprobes are three orders of magnitude smaller than the smallest catheter used in the clinic today. This is incorrect. The smallest catheter in clinical practice is 1.3 French with inner diameter around 300 microns. The magnetic heads of the device described in the manuscript measures up to 350 microns. In addition, the microwires used in clinical practice can be as small as 200 microns.
2. The clinical significance of having such small endovascular probes is unclear; why does the author feel that distal small branches are necessary to access using an endovascular approach? What clinical problems are being envisioned will be solved by achieving this? The smaller the vessels one catheterizes, the greater the danger of a stroke. Small cerebral vessels are also extremely susceptible to vasospasm.
3. The "work" during such procedures are performed by the catheter, which is a conduit to deliver a device such as a stent, aspiration type catheter, angioplasty balloon, etc. The guidewire allows access to the region of interest; then, over this guidewire, the catheter is delivered to the target. Today we have exceptional wires that can traverse many desired blood vessels; however, if the catheter does not track over this wire, as in the tortuous blood vessels displayed in figure 2 and 3, then the procedure would be considered a failure. Once this magnetic probe reaches its target, do the authors envision that catheters would track over this wire to reach the same target? One major challenge today is that these wires are very floppy and soft to minimize injury during navigation; however, the greater the floppiness, the lower the trackability of the catheter especially at sharp angle turns.
4. How will anti-fouling be addressed? In the brain, cannot have any thrombus formation as this will lead to a stroke. Will this system generate heat? Heat is a major promoter of clot formation. The experiments could have been performed using blood. These could be purchased (i.e., pig blood).
5. Navigation of catheters, wires in the brain occur during real-time imaging using x-ray fluoroscopy. In the design proposed, magnetic fields will need to be applied during real-time fluoroscopy to avoid potentially fatal injury to blood vessels; is this possible? For example, applying fields in inaccurate angles could potentially force the magnet at the tip to perforate the artery. Fluoroscopy provides 2D images of a 3D structure; but the magnetic fields may need to be applied along the 3D dimensions. What safeguards would be in place to avoid intima injury, dissection, or thrombosis of these vessels?
6. Instead of using ex vivo animals, the experiments could have been performed in vivo in rabbits. This is a missed opportunity.
7. The discussion begins with the "navigation problem." Again, this needs to be clarified. I am still uncertain what the issues are that requires access to small vasculatures. In addition, the target vessels are also undefined. Given the size of the device that is being navigated, the target is >100 microns? It would help to clarify this as well. Typically for embolization, we avoid using small particles; the smaller the particle the greater the chances for ischemic necrosis, which can then lead to infection, etc..
8. In addition, the discussion should include how the technology described in the manuscript is different from those described in the literature. For example, Kim, Y., Yuk, H., Zhao, R., Chester, S. A. & Zhao, X. Printing ferromagnetic domains for untethered fast-transforming soft materials. *Nature* 558, 274–279 (2018)

Reviewer #2 (Remarks to the Author):

In this manuscript, the author designed an endovascular microrobotic toolkit (tethered ultra-flexible endovascular microscopic probes, μ probes) with a cross-sectional area that is approximately three orders of magnitude smaller than the smallest microcatheter currently available. The size of the μ probes allows

injection with needles smaller than 24G at locations very close to the target areas, minimizing patient discomfort, post-operation complications, and recovery time. This μ probes can be transported through tortuous vascular networks almost effortlessly by harnessing hydrokinetic energy. Dynamic steering at bifurcations is performed by deformation of the probe head using magnetic actuation. However, there are several questions that we find in the manuscript, which are listed as follow :

1. This μ probe device is designed as a tiny size to achieve the fluidity of blood flow and magnetic control, but this design has greater limitations on clinical application. For example, small device lead to a much smaller lumen, and it will be very difficult to inject contrast agents, drugs, embolic agent. The delivery of spring coil and stent is also hard to achieve. Interventional neuroradiology mainly focuses on treatment, and the treatment is mainly carried out through the catheter, which requires a certain caliber to achieve the injection of contrast agents, drugs and embolic agents, the release of devices such as stent and spring coil. Although small caliber probe can increase the flexibility and navigation performance, it will be a big problem how to transport larger caliber treatment catheter along the probe. The probe guide wire is required to have a certain diameter and hardness, which limits the application of this study.
2. The floating character in blood flow of μ probe device is beneficial to the placement of the device, but the uncontrollability in the follow-up treatment will increase the possibility of related complications. A few seconds after perfect placement, the device might move to an unexpected position along with the change of blood flow, which could be fatal risk for the medical treatment.
3. Human skull with large tissue could thickness and depth. The human craniocerebrum has skull with a large thickness and depth of tissue, which is a great challenge for magnetic penetration. In practical clinical application, how to achieve effective magnetic navigation?
4. Many interventional materials such as stent and spring coil are metal products. Since the μ probe was made by metal, how to solve the related interference and safety problems of magnetic navigation during operation?
5. In the main body, the author used many 3D printing models and rabbit ear artery in vitro and ex vivo. These models were mostly two-dimensional structure. However, the human intracranial vessels are three-dimensional structures. We suggest that the author consider 3D printing technology to simulate the human lumen in order to evaluate the feasibility in vitro and in vivo.
6. When the μ probe enters along the blood vessel, it can follow the blood flow. But if the branch blood vessel is completely occluded, there will be no or near no blood flow, how can the device move forward.
7. The μ probe is based on harnessing hydrokinetic energy and using magnetic actuation, and conventional approach relies on a finite proximal force, and the contact pressure exerted on the outward wall to transmit forces distally. According to Supplementary Fig. 1, the tip of the μ probe followed almost identical trajectories while it was being pulled forward by the flow or pulled back by the insertion device. However, when pulled back, the conventional approach could release the force and reduce the damage to the vessel, the μ probe might do damage to the inner wall by the force of onward blood flow. The force analysis (including the friction at corners with high curvature) might be needed to convince the safety of μ probe during the guiding process and retraction process as compared to conventional approach.
8. How to realize the direction of external magnetic field? It seems that the direction of μ probe could be guided only by changing the magnitude of the magnetic field. But can the probe be guided by changing the direction of the magnetic field? What's the difference between the two modes? The author needs more explanation.
9. We know that μ probe can realize controlled-lift and controlled-heading in two-dimensional plane in unidirectional magnetic field environment. We wonder whether does the μ probe need more complex magnetic field control (magnetic field with different orientation) or a unidirectional magnetic field control in a three-dimensional environment? The author needs more demonstrations.
10. The author mentioned that the technology enabled the simultaneous deployment of multiple μ probes. Since the guidance depends on the magnetism, when multiple μ probes are introduced, will the magnetic field affect other μ probes already arranged?
11. In Fig.6a, the flow sensing μ probe was captured to show the heater and sensor elements along with the magnetic head. But our readership may not know the which part of photo corresponds to which part of the μ probe. A diagram might be needed to show the corresponding relationship between the head of probe and heater, sensor element.
12. What are the functions of the heater and sensor? Heating in blood vessels could cause damage on blood vessels, which could mislead our readers. Further clarification might be needed.

13. There was no dense bone effect in rabbit ears. However, if the device is applied to the intracranial, the high density of the skull has an impact on the visibility of the μ probe. DSA images of the μ probe in skeletal environment should be needed to show the application value of the device.

14. The animal experiment was performed on ex vivo Rex rabbit ear. Why not use an in vivo rabbit ear? It could be more convictive if femoral artery puncture was performed and then μ probe was applied in vivo. Besides, other arteries could also be selected to show the three-dimensional state after the application of μ probe.

15. As we know, the internal surface of the blood vessel is smooth, which is conducive to the sliding of the guide wire and the probe. The 3D printing sacrificial negative mould was fabricated and used as an vessel simulation. So, the inner surface of mould might be described and tested to convince it.

16. As the μ probe material is used in the artery, its anticoagulant properties also need to be tested. Overall, we treasure the efforts the authors made, but we have to point out the deficiency as listed above. So, we suggest the manuscript be reconsidered after the authors clarify the questions listed above.

Reviewer #3 (Remarks to the Author):

The paper reports the flexible endovascular microprobes which were navigated by the flow-driven transportation and magnetic steering. They fabricated the microprobes using the laser-machined thin polyimide substrate attached to the magnetic head. The strongest aspect of the manuscript is the characterization of the advancement of the microprobes through the flow-driven and magnetic navigation. The authors also provide some data to suggest the feasibility of the microprobes for flow rate measurements and ex vivo drug injection. However, in my opinion, the idea does not provide any new significant innovations in the endovascular microprobes. Especially, their navigation strategies are limited due to the dimensions and size of the microprobe. Overall, the manuscript remains unclear points that need to be addressed to improve the scientific rigor, and I would recommend this paper to be submitted elsewhere.

Major comments:

1. How can authors navigate the probes when the diameter of the target vessel is similar with the diameter of the magnetic head? If so, the size of the target vessel is too small to generate the drag force and hydrodynamic lift due to the low velocity gradient. Furthermore, the magnetic head cannot be rotated because of the confined inner lumen of the vessel.
2. Does the microprobe still advance when there are vortices in a fluid in which the flow revolves around a streamline? The vortices could be produced in arterial branches or the blood vessels with the stenosis, which can interrupt the flow-driven navigation of the probes.
3. There should be more information in the 'Magnetic actuation of the tip for controlled bending of the microprobe' section. The authors seem to clamp the proximal tip of the microprobe during the characterizations of the advancement and the navigation strategy. However, the magnetic torque could be changed by the distance between the clamped end and the magnetic head. For example, if the probe is inserted further, the deflection angle of the magnet heads and the probe body will be increased in an external magnetic field compared to the microprobes with shorter insertion depth. Furthermore, when the probe contacts with the wall, the magnetic torque also can be changed because the contact point acts as a new clamped end. The author should provide some estimate for such cases.
4. How reliable is the magnetic head? The magnetic head was composed of the mixture of the PDMS and NdFeB microparticles. The NdFeB alone is not biocompatible. Furthermore, the water can be absorbed into the PDMS and cause the corrosion of the NdFeB, which produces cytotoxic byproducts. The author should evaluate the biocompatibility and reliability of the device.

Minor comments:

1. In figure 2a, what are the radius of the curvature of the second and third turns? In figure 2b, why did the probe buckle? If the probe advances along with the centerline of the vessel, the probe body cannot be bended until it touches the wall. In figure 2d, please provide the unit and value of the vertical axis.
2. In ex-vivo demonstration, how did you quantify the amount of the delivered drug at the target vessels? For example, in the case of figure 3e, the flow rates of the drug in each vessel branch will be different due to the difference in the flow resistance based on the diameter of the vessel.

REVIEWER COMMENTS

Reviewer #1 (Remarks to the Author):

In this manuscript, tethered ultra-flexible endovascular micro-probes that are three orders of magnitude smaller than the smallest microcatheter are developed. These probes can be steered and guided to areas of the vascular network otherwise not accessible by conventional catheters used in the clinic today. While this is very exciting with great results, I have several concerns that could be addressed in the introduction or in the discussion.

We thank the reviewer for his/her enthusiastic support for publication of this work and have strived to address his/her concerns in the revised manuscript.

Specific Comments

1. In the introduction it is said that the microprobes are three orders of magnitude smaller than the smallest catheter used in the clinic today. This is incorrect. The smallest catheter in clinical practice is 1.3 French with inner diameter around 300 microns. The magnetic heads of the device described in the manuscript measures up to 350 microns. In addition, the microwires used in clinical practice can be as small as 200 microns.

Authors' Response: We thank the reviewer for pointing out the confounding measurement units found in the field of medical devices and giving us the opportunity to better define the specificities of our technology.

The smallest catheter used in neurological interventions is the Magic 1.2 French (Balt Extrusion, Montmorency, France) with an outer diameter of 400 μm . The smallest guidewire commercially available that fits through this catheter is the Hybrid 0.007-inch (Balt Extrusion, Montmorency, France), which corresponds to a diameter of 178 μm . Bending stiffness of the device is the key element for the flow-driven navigation. To this end, we compared the cross-sectional area of the body of our μprobes , and not the head, with that of the commercially available catheters. The largest device that we used was the flow sensor shown in Figure 6, which measured 4 μm by 300 μm (thickness x width) leading to a cross-sectional area of 1,200 μm^2 . The smallest μprobe used in this work had

dimensions of 4 μm by 25 μm (thickness x width), with a cross-sectional area of 100 μm^2 (Supplementary Figure S9). The cross-sectional area of a 1.2 French catheter is 126,000 μm^2 . From these numbers, we can see that the difference is indeed three orders of magnitude. The magnetic head can be as thin as the body if we use hard magnetic thin films, which would give the same effective magnetization as dispersing magnetic microparticles in an elastomer matrix. These films can be prepared using laser cutting or electrodeposition (see **Figure R1**).

Figure R1. Microfabrication process for ultra-small μprobes (25 μm -wide and 4 μm -thick) with metallic $25 \times 25 \times 25 \mu\text{m}^3$ magnetic heads. **a** The magnetic thin film is laser cut to produce the head with desired geometry and size, which is then glued to the distal tip of the μprobe . **b** Microscope images of the μprobe showing the laser-cut magnetic structure. Scale bar, 50 μm .

2. The clinical significance of having such small endovascular probes is unclear; why does the author feel that distal small branches are necessary to access using an endovascular approach? What clinical problems are being envisioned will be solved by achieving this? The smaller the vessels one catheterizes, the greater the danger of a stroke. Small cerebral vessels are also extremely susceptible to vasospasm.

Authors' Response: The reviewer raises two important questions. Concerning the ability to reach very peripheral vessels or to access those that are currently too small to be catheterized: this might allow endovascular specialists to penetrate the so-called perforating arteries, thereby giving access to therapeutic options in structures such as the brain stem, the retinal arteries or the basal ganglia. Such access might open new therapeutic options to treat deep-seated or very peripheral tumors inside the brain, at the skull base or alongside the meningeal lining but also target movement or epileptic disorders or thromboembolic diseases by allowing drug injections, neural stimulation or recording in vascular zones currently unattainable. Moreover, the concept of transporting multiple μ probes with the flow will allow easier penetration in the depths of arterio-venous malformations and may increase the obliteration rate using liquid embolic agents while limiting the risk of excessive embolization of arterial feeders giving off normal branches to the surrounding brain parenchyma. The same holds true for dural fistulas that are often fed by very tortuous feeders (typically the posterior or middle meningeal arteries) that sometimes prove to be impossible to cannulate distally at a satisfactory level for a thorough and safe obliteration of the shunting zone. Other delicate structures that could definitely benefit from enhanced atraumatic peripheral navigation are the spinal arteries that are known to be fragile and dangerous to catheterize.

Regarding the second question, since our method of navigation does not require any mechanical push and does not put the vessel wall under tension or deform the arteries, the risk of spasm is expected to be much lower than what one may see in current clinical practice during conventional navigation with a catheter over a guidewire. This was observed first-hand by one of the co-authors who is an experienced endovascular neurosurgeon. Please also see our response to the 7th comment of the second reviewer and **Supplementary Note 5**. The risk of vasospasm may theoretically increase during the retraction of the μ probes compared to the advancement phase but it is known that mechanically-induced vasospasms resolve spontaneously after removal of the material or administration of anticalcic drugs (calcium antagonists). This is typically a feature that we will study in our future animal trials.

We added a paragraph to the Discussion to elaborate the clinical significance of the presented technology. A complete force-damage analysis is incorporated into the Supplementary Information as Supplementary Note 5.

3. The “work” during such procedures are performed by the catheter, which is a conduit to deliver a device such as a stent, aspiration type catheter, angioplasty balloon, etc. The guidewire allows access to the region of interest; then, over this guidewire, the catheter is delivered to the target. Today we have exceptional wires that can traverse many desired blood vessels; however, if the catheter does not track over this wire, as in the tortuous blood vessels displayed in figure 2 and 3, then the procedure would be considered a failure. Once this magnetic probe reaches its target, do the authors envision that catheters would track over this wire to reach the same target? One major challenge today is that these wires are very floppy and soft to minimize injury during navigation; however, the greater the floppiness, the lower the trackability of the catheter especially at sharp angle turns.

Authors’ Response: We are glad that the reviewer raised these important technical aspects, giving us the opportunity to clarify the unique features of our device. Our μ probe is definitely not meant to replace a guidewire and, hence, not intended for tracking a catheter over it. As opposed to current navigation tools that have to be actively pushed forward, we offer an alternative technique where the material is passively carried by the stream and actively held or released like a kite in the wind by modulating the magnetic torque at the tip. Conventional endovascular interventions rely on the synergistic activity of two different structures: guidewire and catheter. The guidewire is inserted into the vessel through pushing (to move forward) and rotating/twisting (also known as “torqueing” to make turns) at the proximal end by the operator. Once the guidewire is in place, the catheter is coaxially slid into the vessel. As the reviewer pointed out, these wires are made to be progressively more floppy distally to minimize injury during navigation; however, the greater the floppiness, the lower the trackability of the catheter especially at sharp angle turns. More importantly, the current guidewire technology has already reached the physical limits of floppiness, meaning that a further size reduction of the existing guidewires will not provide trackability for catheters due to mechanical instabilities. In our approach, we abandon this two-step navigation paradigm. The integrated microfabricated device, that essentially serves as both guidewire and catheter, gets naturally carried by the blood flow to reach distal arterioles that would otherwise remain inaccessible. This type of advancement is not possible with commercially available guidewires or the so-called flow-related microcatheters due to their structural and geometric properties.

Aside from its effortless and atraumatic navigation to the far depths or periphery of the vascular system, incorporation of microfluidic, electronic, and electromagnetic components is meant to enable injection of therapeutic agents, measurement and manipulation of local mechanical microenvironment, and/or neurostimulation or recording. As the microfabricated μ probes are significantly smaller than the standard catheters, navigation capabilities may be different and procedures must be performed using different physical principles. For example, inertial effects are negligible at small scale while viscous stresses become predominant. Heat transport happens almost instantaneously and surface effects such as adhesion or van der Waals interactions apply relatively large forces. The way the μ probes will perform during injection, embolization, cauterization, measurement of mechanical properties, recording of electrical activity, and neurostimulation will mostly differ from conventional techniques due to these physical scaling effects.

We clarified these points in the Introduction and Discussion sections of the revised manuscript.

4. How will anti-fouling be addressed? In the brain, cannot have any thrombus formation as this will lead to a stroke. Will this system generate heat? Heat is a major promoter of clot formation. The experiments could have been performed using blood. These could be purchased (i.e., pig blood).

Authors' Response: We thank the reviewer for raising these important issues. Clot formation must be avoided at all costs. The materials that we use for the manufacturing of the μ probes have been extensively used for the development of implantable biomedical devices. More specifically, previous work has shown that devices made out of Polyimide or PDMS polymers did not display thrombogenicity, biofouling, or cytotoxicity in vivo (Starr et al., J. Biomed. Mater. Res., 2016; Rajan et al., Int. J. Biomater., 2013; Richardson et al., Biomaterials, 1993; Constantin et al., Materials, 2019) and can be chemically treated to improve their compatibility with the host tissue and lower their friction coefficient (Leslie et al., Nat. Biotechnol., 2014; Epstein et al., PNAS, 2012; Trel'ová et al., Langmuir, 2018; Kim et al. Science Robotics, 2019). Notably, during our ex vivo experiments, no macroscopic lesions or vessel sclerosis was observed in the rabbit ears used for navigation and ink injection.

The navigation paradigm is not dependent on the material choice. We can tune the bending stiffness of microfabricated devices by changing their design so that flow-driven navigation would work. We demonstrated this versatility by manufacturing devices from a wide range of stiffnesses, PDMS (Young's modulus: 10 MPa) and Kapton (Young's modulus: 4 GPa). Devices can be fabricated from medical-grade FDA approved polymers that are used to manufacture commercially available catheters such as polyurethane (Young's modulus: 10 – 900 MPa) and polyethylene (Young's modulus: 100 – 800 MPa). The design paradigm can also be adapted to stiffer materials such as glass, metal and ceramics, or hydrogels such as gelatin, opening up the floor for biodegradable electronic devices (Baumgartner et al, Nat. Mater., 2020).

We evaluated the anticoagulant properties of our μ probes following a standard ISO-10993-4 protocol. We used fresh human blood and measured the ATP release from platelets (platelet activation) and the platelet aggregation, which are crucial steps in the blood clotting process. As shown in **Figure R2**, the μ probes did not induce neither platelet activation nor platelet aggregation, revealing that the polymers that we carefully chose do not induce thrombus formation in their original finishing. The surface properties of the μ probes may be modified if needed using an additional bioinspired omniphobic surface coating, that was previously shown to drastically enhance anti-fouling and anti-coagulation properties of medical devices (Leslie et al., Nat. Biotechnol., 2014). The omniphobic coating is achieved by covalently bonding a tethered perfluorocarbon layer to the surface, on which an FDA-approved mobile layer of perfluorodecaline layer is deposited. The protocol is compatible with a variety of biomedical polymers, including polyimide and PDMS.

Figure R2. Blood compatibility assay performed on fresh human blood under high shear stress condition. Three different samples were tested, kapton film (K: 0.004 x 0.2 x 10 mm³), magnetic

head with silica-coated magnetic particles (H: \varnothing 0.3 mm x 1 mm), and whole μ probe (P: K + H). Negative and positive control are platelet rich plasma (PRP – non-activated platelets) and PRP exposed to collagen type-I (PRP+Col-I – activated platelets). **a** ATP release from platelets was measured using luciferin-luciferase reaction. **b** Platelet aggregation was measured using absorbance (595 nm). The aggregation % was defined by the transmission of light through non-activated PRP, represented as 0 % aggregation, and through platelet-poor plasma (PPP) as 100 % aggregation as previously described by several groups. Both techniques have been vastly used in the platelet field to investigate the thrombogenicity of different materials/devices. Values were compared with the non-activated platelet, (**P < 0.001, PRP) by one-way ANOVA followed by Dunnett's post hoc test (SEM, duplicate average from four independent experiments).

Neither the fluid-driven navigation nor the magnetic actuation of the head generates heat. As a comparison, conventional magnetic resonance imaging (MRI) machines continuously apply magnetic fields up to 7 T for minutes to hours with no reported heating effects. Only the flow sensor generates heat during the brief (millisecond) application of the electrical pulse. We developed a computational model of the system and show that the temperature is only raising at the very surface of the μ probe due to the shear flow and it stays within safe conditions. Please see our response to the 12th comment of the second reviewer for a detailed analysis.

The anticoagulation study is summarized in the new subsection “The translation of the technology to in vivo applications” and the results are shown in Supplementary Figure 16. The anti-fouling property is discussed in the same subsection.

5. Navigation of catheters, wires in the brain occur during real-time imaging using x-ray fluoroscopy. In the design proposed, magnetic fields will need to be applied during real-time fluoroscopy to avoid potentially fatal injury to blood vessels; is this possible? For example, applying fields in inaccurate angles could potentially force the magnet at the tip to perforate the artery. Fluoroscopy provides 2D images of a 3D structure; but the magnetic fields may need to be applied along the 3D dimensions. What safeguards would be in place to avoid intima injury, dissection, or thrombosis of these vessels?

Authors' Response: There are two important questions in this comment. 1. Can we localize the μ probes using x-ray fluoroscopy during operation? 2. What would happen if we apply magnetic fields in the wrong direction? Is there a risk of perforation?

Regarding the first question, as the reviewer pointed out, localization is an important aspect of navigation. We believe that our technology can be adapted to perform fully automated navigation using a map of the vasculature, created with pre-operative medical imaging. However, it is important to have the ability to detect the position of the head of the μ probe during the surgery to give the control to the surgeon for teleoperated interventions and for safety reasons. Furthermore, it may not be always possible to acquire a map of the vasculature prior to the operation.

We verified visibility (radio-opacity) in fluoroscopy using biplane angiographic suites in partnership with radiology technicians and an experienced endovascular specialist. We prepared samples (**Figure R3a,b**) with a series of head geometries, and recorded fluoroscopic and radiography images using two different fluoroscopy systems. The first system is a clinical fluoroscope for head interventions (Alphenix Sky+, Canon Medical Systems Corporation) that is currently being used in the operating room at the Lausanne University Hospital (CHUV). The system was able to resolve magnetic heads as small as 200 μ m in diameter with low radiation-dose fluoroscopic images through a rabbit head as a phantom (**Figure R3c**). The relatively poor detection capability of features below 200 μ m is due to the intrinsic resolution of the imaging machine, which is 200 μ m/pixel. The second system is a more advanced clinical fluoroscope (Alphenix Core+, Canon Medical Systems Corporation) with 76 μ m/pixel high-definition resolution. Here, the smallest feature clearly detectable with low radiation-dose fluoroscopy imaging inside a commercially available head phantom was 150 μ m (**Figure R3d**). In both fluoroscope systems, live imaging allowed the eye to spot smaller features with respect to single fluoroscope images. Moreover, higher dose radiography images allowed to detect smaller features (150 μ m for the Alphenix Sky+ and 75 μ m for the Alphenix Core+), which is going to be useful for sparse and random localizations of the μ probe in potential automated tests without ramping up the total absorbed dose. Finally, magnetic actuation is compatible with fluoroscopy as magnetic fields do not interfere with x-rays. See **Movie R1** for a demonstration.

Figure R3. Imaging of μ probes using two different fluoroscopes in realistic environments. **a,b** Optical images of the samples under test. The composite consists of PDMS:NdFeB microparticles that was mixed at a 1:1 volume ratio. The diameter of the magnetic samples varied between 150 μ m and 350 μ m. **c** Fluoroscope images (Canon Alphenix Sky+, 200 μ m resolution) of the magnetic structures (top) placed in an ex vivo rabbit head (bottom). **d** Fluoroscope images (Canon Alphenix Core+, 76 μ m resolution) of the magnetic structures (top) inside an anthropomorphic head phantom (bottom). Scale bars, 5 mm.

These results prove that our technology can be translated into clinics with existing imaging systems for μ probes down to 150 μ m in size. X-ray fluoroscopy technology with a focal spot size of 10 μ m already exists (Glenbrook Technologies) and could thus be implemented in machine updates in the near future. We believe that current clinical imaging modalities are not tailored for imaging μ probes not because of fundamental scaling limitations but to keep the field of view large for large tools. As a reminder, X-ray computer tomograph systems can resolve sub- μ m features. There are two potential

solutions for increasing the resolution of x-ray imaging, i) using more advanced fluoroscope systems tailored for imaging smaller structures and ii) incorporating contrast agents such as bismuth subcarbonate, bismuth oxide or barium sulfate into the magnetic head.

Regarding the second question, safety issues are of paramount importance and we thank the reviewer for giving us the opportunity to clarify this point. The navigation principle of our μ probe is almost the opposite as the traditional push over the wire method used nowadays. Even the so-called flow dependent microcatheters such as the Magic (Balt, USA) need to be pushed despite the fact that saline or contrast injection allows experienced operators to select different bifurcation branches without a microwire. Some form of forward tension by applying push is nonetheless necessary and most operators tend to use a guidewire at some point, especially for very distal branches when trying to reach deep feeders such as in AVM embolization. The μ probe that we have designed is dragged forward by the blood flow in the same manner as a kite in the wind, with the difference that our drag element is the entire μ probe. The properties of its tip allow it to be held stationary in the flow by the magnetic field or redirected in one direction or the other but always in the stream of the blood flow. The μ probe comes into contact with the blood vessel only at curved corners.

We ran additional navigation trials in agarose molds to empirically show that the forces acting on the walls are insufficient to cause damage. Agarose phantoms are widely used to mimic brain tissue as they display the same consistency and resistance to mechanical damage as brain parenchyma. Unlike standard guidewires, we could not detect any damage (perforation, dissection, etc.) on the vessel walls of relatively long and tortuous vasculature during advancement or retraction of μ probes. Please see our response to the 7th comment of the second reviewer for a detailed analysis. This was also the case for the ex vivo studies in the rabbit ear where no perforation was observed, even when purposely tried.

To be more quantitative, we analytically calculated the forces that are applied by the magnetic head due to application of magnetic fields using an experimentally validated model. We do not apply push or pull forces to the magnetic head because we use homogeneous magnetic fields for steering. The only force exerted by the magnetic head on vessel wall arise from the magnetic torque and the rotation of the head (**Figure R4**).

These forces are transmitted at the longitudinal extremes of the head and are maximized with the application of a perpendicular magnetic field. We assumed the worst-case scenario in the following analysis to report the highest possible forces.

Figure R4. Illustration showing the free body diagram of the magnetic head. Contact force was applied by the tip of the magnetic head on the vessel wall upon application of magnetic torque.

The total force acting at the tip can be computed from the torque – force relation assuming perpendicular actuation (i.e. magnetic field is perpendicular to the longitudinal axis of the probe head):

$$F = \frac{M}{L}$$

where M is the torque applied by the magnetic field and L the distance between the center of rotation and the tip. The magnetic torque is computed from:

$$\mathbf{M} = \mathbf{m} \times \mathbf{B}$$

where m is the magnetic dipole moment and B the applied magnetic field. The vector product imposes maximal torque at 90° between applied magnetic field and magnetization direction of the magnetic head ($\mathbf{B} \perp \mathbf{m}$). The magnetic dipole moment can be computed as:

$$m = \frac{1}{\mu_0} B_r V$$

where μ_0 is the permeability of vacuum ($= 4\pi \cdot 10^{-7} \text{ H}\cdot\text{m}^{-1}$), B_r is the factory-measured residual flux density of the neodymium iron boron (NdFeB) magnetic particles (0.8 T) and V is the volume of the magnetic head (20% volumetric ratio between NdFeB particles and PDMS).

Next, we calculated the maximal force that the magnetic head would exert on the vessel wall if we accidentally applied the magnetic field in the most misdirected way. The maximum field strength that we used for steering is 30 mT and the largest magnetic head has a diameter of 350 μm . For these parameters, the maximal force was calculated to be around 0.4 mN (**Figure R5**).

Figure R5. Calculation of the magnitude of the contact force applied by the tip of the magnetic head on the vessel wall. The data is plotted as a function of the head diameter. Magnetic field is assumed to be perpendicular to the long axis of the head, at a constant magnitude of 30 mT.

We experimentally validated the analytical model by measuring the change in the weight of an actuated magnetic head (**Figure R6**). To avoid magnetic interaction between the Helmholtz coils and the scale, a light plastic bar was vertically positioned on the scale to transmit the forces applied from the fiber. We applied uniform magnetic field of 20 mT strength to a horizontally placed 5 mm long and 700 μm thick magnetic fiber that was clamped at one end. For these parameters, the calculated force is 0.98 mN. The increase in mass that was measured by the scale was 0.09 g, which corresponds to 0.88 mN.

Figure R6. Schematics of the setup used to measure the force applied by the tip of the actuated magnetic head. The setup consists of a pair of electromagnetic coils (Helmholtz configuration) placed above a sensitive scale. A lightweight bar serves as an adapter to transmit the forces generated by the clamped magnetic head that is kept in position by a fixed holder (top inset). Upon application of homogeneous magnetic field perpendicular to the long axis of the head, the structure rotates and presses on the vertical bar (bottom inset).

Unfortunately, we could not find values of puncture forces in the literature on endovascular catheters. In a related work, the puncture forces for microfabricated beveled microneedles were measured in retinal veins using a sensitive load cell (Ergeneman et al. *J. Med. Devices*, 2011). The similarity of the size of the vessels to that of the targeted cerebral vasculature allows a fair comparison. At a comparable sub-mN force range, the tip size of the needles must be at least 10 times smaller than the size of the magnetic head (Figure R5). Likewise, reducing the size of the magnetic head to $30 \mu\text{m}$ would decrease the maximal force to $2.7 \mu\text{N}$ at $B = 30 \text{ mT}$, which is at least 2 orders of magnitude smaller than the force required for a $30 \mu\text{m}$ microneedle to puncture retinal veins. Notably, this comparison was made against beveled needles explicitly designed to decrease puncture force. We can further tune the design of the head to provide perfectly smooth edges (i.e. increase contact surface), fabricate them from extremely soft materials (e.g. Ecoflex with modulus in the kPa range), and coat them with biocompatible lubrication materials to further mitigate perforation and dissection risks.

The results of fluoroscope imaging is summarized in the new subsection “The translation of the technology to in vivo applications” and shown in Supplementary Figure 13. A complete force-damage analysis is presented in Supplementary Note 5.

6. Instead of using ex vivo animals, the experiments could have been performed in vivo in rabbits. This is a missed opportunity.

Authors’ Response: Although we agree with the reviewer, the legal authorities and ethical standards in our country require extensive in vitro and ex vivo testing before any animal trial can be started (3R principle: Replace, Reduce, Refine). Our main objective was to establish a proof-of-concept to pave the way towards such in vivo trials. We believe we showed enough evidence to claim that the materials and methods are applicable to clinical settings. We are confident that such experiments will be supported once our preliminary work becomes publicly available.

7. The discussion begins with the “navigation problem.” Again, this needs to be clarified. I am still uncertain what the issues are that requires access to small vasculatures. In addition, the target vessels are also undefined. Given the size of the device that is being navigated, the target is >100 microns? It would help to clarify this as well. Typically for embolization, we avoid using small particles; the smaller the particle the greater the chances for ischemic necrosis, which can then lead to infection, etc..

Authors’ Response: Our flow-driven μ probes could be used to reach the depth of lesions currently unattainable, such as hypervascularized tumors of the calvarium or skull base, as well as pial or dural arteriovenous malformations and fistulas of the brain or spine that are often too tortuous or delicate to be navigated as distally as anticipated due to increasing tension and vasospasm. Another possible application could be neuromonitoring or modulation of movement related disorders of the basal ganglia or brain stem structures through perforating arteries. Currently we are able to design μ probes as small as 50 μ m. Given that the perforating lenticulostriate branches of the middle cerebral artery measure at least 80 μ m (Marikovic et al, Stroke, 1985) we can expect to reach these vessels and perhaps the most peripheral terminal branches measuring as little as 50 μ m. The main limitation for clinical practice would be the resolution and magnification of the

angiographic suites currently used in clinical practice for proper visualization. Please see our response to the 5th comment of the reviewer for a detailed discussion on this topic.

As the reviewer pointed out, we anticipate to experience negative implications of scaling in mass transfer. On the other hand, we believe that miniaturization will bring new technological opportunities for accomplishing the same tasks. Please see our answer to the 3rd comment of the reviewer. While some form of liquid embolization could be a possible application (depending on viscosity and microfluidics characteristics), we do not anticipate particle embolization to be compatible with our system. Endovascular sclerotherapy, however, might represent an interesting therapeutic application of our μ probe. Indeed, we can locally generate enough heat to collapse the vessels using Joule heating. Although we used low power pulses for the flow measurements, high temperatures can be achieved at higher currents and longer duty cycles or wireless inductive heating. Endovascular cauterization in small distal and curvy vessels through magnetic-heating of the tip could be used for rescue therapy in case of hemorrhagic perforation or for segmental occlusion of AVM feeders. Likewise, previous work hinted on the potential of using a vibrating catheter tip for ultrasound thrombolysis and vasodilation (Hamm et al., *Am. J. Cardiol.*, 1997). Using ultrasound energy is expected to become more effective with decreasing vessel size because RF heating depends on the volume of the magnetic material. Previous work has shown that piezoelectric transducers can be miniaturized to the size of our μ probes (Han et al., *Nat. Electron.*, 2019; Dagdeviren et al., *Extreme Mech. Lett.*, 2016). The capacitive actuation of piezoelectric transducers (high voltage, low current) makes it a favorable actuating scheme for small electronic devices. Finally, implementation of a selective detachment of μ probe components for embolic purposes such as aneurysms is another route that we are exploring.

We added a paragraph to the Discussion to elaborate the clinical significance of the presented technology.

8. In addition, the discussion should include how the technology described in the manuscript is different from those described in the literature. For example, Kim, Y., Yuk, H., Zhao, R., Chester, S. A. & Zhao, X. Printing ferromagnetic domains for untethered fast-transforming soft materials. *Nature* 558, 274–279 (2018)

Authors' Response: As mentioned in the introduction, previous work showed that bending the tip of elastic catheters using magnetic fields facilitates their deployment. However, these techniques still rely on the pushing of the catheter, which faces a fundamental mechanical limitation with decreasing size or elasticity. For example, in the most recent work from the same group that is more relevant to this discussion (Kim et al., *Sci. Robot.*, 2019), it was shown that increasing deflection capabilities by decreasing stiffness increases the number of contact points with the phantom walls in order to sustain the contact-slide based navigation strategy. This confirms that decreasing the stiffness further has serious limitation in scalability and will ultimately fail because of mechanical instabilities.

By harnessing flow, we discovered a way to bypass a fundamental tradeoff in front of the miniaturization quest: trackability vs. pushability. Our findings revealed several additional advantages such as significantly reduced deployment along with inherent safety. Another important limitation of the existing technology such as the work from Kim et al. is the incorporation of electronics. In the presented prototypes, the whole device was casted or extruded from magnetic elastomers which makes it quite challenging to incorporate electronic components. Not surprisingly, all these devices are shaped in the form of a tube, and only used for injection of chemicals or as a lightguide. On the other hand, we engineered the body of our μ probes from a polymer (polyimide) that can be processed using conventional clean-room techniques, which are readily integrated with printed circuit board and semiconductor technology. We can easily pattern electrical and electronic circuit components during the fabrication of the μ probe, as demonstrated by the flow sensor. This way, we leverage the recent advances in flexible and injectable electronics. The door is open for incorporating CMOS architectures, on-board mechanical actuators, 3D folded structures.

Reviewer #2 (Remarks to the Author):

In this manuscript, the author designed an endovascular microrobotic toolkit (tethered ultra-flexible endovascular microscopic probes, μ probes) with a cross-sectional area that is approximately three orders of magnitude smaller than the smallest microcatheter currently available. The size of the μ probes allows injection with needles smaller than 24G at locations very close to the target areas, minimizing patient discomfort, post-operation complications, and recovery time. This μ probes can be transported through tortuous vascular networks almost effortlessly by harnessing hydrokinetic energy. Dynamic steering at bifurcations is performed by deformation of the probe head using magnetic actuation. Overall, we treasure the efforts the authors made, but we have to point out the deficiency as listed below. So, we suggest the manuscript be reconsidered after the authors clarify the questions listed below.

We are pleased the reviewer found the approach and results promising and hope that the following explanations and additional experiments provide adequate clarification.

1. This μ probe device is designed as a tiny size to achieve the fluidity of blood flow and magnetic control, but this design has greater limitations on clinical application. For example, small device lead to a much smaller lumen, and it will be very difficult to inject contrast agents, drugs, embolic agent. The delivery of spring coil and stent is also hard to achieve. Interventional neuroradiology mainly focuses on treatment, and the treatment is mainly carried out through the catheter, which requires a certain caliber to achieve the injection of contrast agents, drugs and embolic agents, the release of devices such as stent and spring coil. Although small caliber probe can increase the flexibility and navigation performance, it will be a big problem how to transport larger caliber treatment catheter along the probe. The probe guide wire is required to have a certain diameter and hardness, which limits the application of this study.

Authors' Response: Please see our responses to the 2nd and 3rd comments of the first reviewer for the clarification of the intended use and the differences between conventional endovascular catheterization and our approach. We added an extended

coverage on this topic to the Introduction and Discussion sections of the revised manuscript.

The current endovascular therapeutics consist of combining catheters and wires to reach a target vessel and to then deliver solid or liquid implants or drugs. Our μ probe is not designed to fulfill these requirements but rather intended to be self-sufficient in terms of navigation and therapeutics. As correctly pointed out, we rely on the ultra-flexibility of the device, thus, unlike conventional catheters, we cannot transport metal wires or coils to the target location. On the other hand, forces that have negligible effects at macroscale may play a dominant role at microscale. For example, fluid flow can pull and transport our μ probes because viscous forces can dominate the elastic forces at this scale. While some form of liquid embolization may be a possible application, it is true that the viscosity of contrast agents or drugs require appropriate pressure conditions. We demonstrated that ink could be injected through our system but obviously this may not necessarily be reproducible in longer hollow μ probes measuring 150-200 mm, especially for more viscous substances. We envision that, instead of the application of pressure, we can harness other forces for mass transport, such as Archimedean screw mechanisms or thermocapillary flows (Huang et al., *Adv. Mater.*, 2015; Zhang et al., *Nature Communications*, 2019). Heat transfer is rapid and localized at microscale and hyperthermia using radio frequency (RF) magnetic fields has been successfully implemented in various vivo settings (Xiang et al., *Polymers*, 2020). To this end, endovascular cauterization through RF heating of the tip, such as rescue therapy in case of hemorrhagic perforation, is one intended application.

On the other hand, we foresee great potential on long-term recording and stimulation of neural tissues through arterial access. Seminal work has shown that these tasks can be performed in cerebral veins using large stent-shaped electrodes (Oxley et al., *Nature*, 2016; Opie et al., *Nat. Biomed. Eng.*, 2018). The ability to simultaneously deploy multiple leads will enable 3D mapping of activity and spatiotemporally controlled activation at multiple sites. Optogenetics emerged as a versatile technique to address numerous scientific questions in neuroscience (Mickle et al., *Nature*, 2019; Montgomery et al., *Nat. Methods*, 2015; Park et al., *Nat. Biotechnol.*, 2015), however, state-of-the-art stimulation devices are bulky and invasive. With our technology, accessing deep brain regions through the endovascular path appears feasible, which would enable surgery-free optogenetic stimulation using designer μ probes. In addition to the animal welfare, the

minimally invasive deployment of devices may enable longitudinal behavioral studies, where stress and constraints must to be minimized.

2. The floating character in blood flow of μ probe device is beneficial to the placement of the device, but the uncontrollability in the follow-up treatment will increase the possibility of related complications. A few seconds after perfect placement, the device might move to an unexpected position alone with the change of blood flow, which could be fatal risk for the medical treatment.

Authors' Response: The characteristics of flow in small vessels (< 2 mm) is to our advantage for position control. The flow is laminar, vortices-free and effectively non-pulsatile at this scale. Thus, the tension due to the viscous drag forces is uniaxial and constant along the longitudinal axis of the μ probe (x-axis in our plots). As long as we keep the length of the μ probe constant, the head of the μ probe remains at the same position along the longitudinal axis. There is no fatal risk in terms of damage to the vessel walls or transport to a different location. Yet, fluctuations in the transversal position of the head (y-axis in our plots) are expected. If high precision in positioning is needed, we can stabilize this transversal motion of the μ probe head through the application of external magnetic fields.

3. Human skull with large tissue could thickness and depth. The human craniocerebrum has skull with a large thickness and depth of tissue, which is a great challenge for magnetic penetration. In practical clinical application, how to achieve effective magnetic navigation?

Authors' Response: We are applying low-strength (up to 30 mT), uniform, and static magnetic fields. Under this regime, we are expected to have the same magnetic field strength everywhere in our workspace, which is defined by the size of the Helmholtz coils, regardless of the thickness and composition of the biological tissue. We will make sure that all the tools are non-magnetic and there are no inductive circuit elements. To demonstrate the invariance of magnetic field with respect to the presence of biological tissues, we measured the magnetic field strength 25 cm away from a cylindrical permanent magnet (diameter = length = 4 cm, HKCM Engineering, Germany) with and without a human head placed in between the magnet and a magnetometer (GM08, Hirst Magnetic

Instruments Ltd., UK). See **Figure R7** for an illustration of the experimental configuration. The field strength was recorded as $125.1 \pm 2.8 \mu\text{T}$ and $126.0 \pm 2.9 \mu\text{T}$ with and without the head, confirming that the presence of the head has no significant statistical effect on the propagation of the magnetic field (two-tail paired t-test, $\alpha = 0.05$, $P = 0.153$, $n = 33$).

Figure R7. Measurement of magnetic field strength in the presence of human skull. Illustration showing the placement of the magnetic field source and the gaussmeter. The magnet and the sensor were placed 25 cm apart across a the head of a human volunteer.

4. Many interventional materials such as stent and spring coil are metal products. Since the μ probe was made by metal, how to solve the related interference and safety problems of magnetic navigation during operation?

Authors' Response: Magnetic materials must be kept away from the workspace for safety reasons. The μ probes have gold/titanium/platinum electrodes and other circuit elements, which show no magnetic response under the fields that we apply for the steering of the probe. The stents or spring coils can be fabricated from non-magnetic materials.

5. In the main body, the author used many 3D printing models and rabbit ear artery in vitro and ex vivo. These models were mostly two-dimensional structure. However, the human intracranial vessels are three-dimensional structures. We suggest that the author consider 3D printing technology to simulate the human lumen in order to evaluate the feasibility in vitro and in vivo.

Authors' Response: We thank the reviewer for this comment. Our electromagnetic control system can generate arbitrary uniform magnetic fields in 3D. We worked with 2.5D

phantoms to be able to quantify the kinematics of the μ probe and fluid flow. To our knowledge, there is no 3D printing technique that allows rapid printing of sub-mm channels with arbitrary complexity in a large workspace and from transparent materials. One contribution of the presented work is the fabrication of such phantoms. We anticipate that our work will inspire companies, such as Elastrat, to develop FDA-approved phantoms with microvasculature adapted to endovascular flow-driven μ devices.

To address the reviewer's comment, we fabricated a phantom with a truly 3D vessel network using our sacrificial molding technique that is based on dissolving of polyvinyl alcohol filaments (see Methods section and Supplementary Figure S19). The main channel (2.7 mm in diameter) branches into three bifurcations at different locations. The diverging vessels (1.7 mm in diameter) were designed to occupy the 3D space and navigation in this phantom requires control of tip orientation in all three cartesian axes. For the visualization of the μ probe motion, we used two cameras observing the workspace from two different sides of the phantom. The μ probe navigated effortlessly against gravity, validating that gravitational forces are negligible compared to viscous forces at small scale (**Figure R8**). See **Movie R2** for a demonstration. For the details of navigation control in 3D vasculature, please see our response to the 7th and 8th comments of the reviewer.

We incorporation these results to the main text (page 10) and showed the data in Supplementary Figures S6 and S7.

Figure R8. Teleoperated navigation in a 3D phantom. **a** The workspace is observed by two cameras (A and B) for 3D localization. The targets are marked with roman numerals i, ii and iii. **b** Snapshots from the navigation movie showing successful deployment of the μ probe at different daughter arterities. Scale bars, 5 mm.

6. When the μ probe enters along the blood vessel, it can follow the blood flow. But if the branch blood vessel is completely occluded, there will be no or near no blood flow, how can the device move forward.

Authors' Response: This issue is indeed the Achilles heel of our technology. If the vessel is completely blocked then flow-driven navigation is not going to work anymore. However, if this is a local and temporary situation and our objective is to perform an operation for opening a blockage (i.e. thrombolysis), we can actuate the head to perform crawling motion to be able to advance further (see Supplementary Note 2). Detailed characterization of this propulsion mode is going to be the focus of our future work.

If the flow is drastically reduced in a channel with a width substantially larger than the head size, we can actuate the structure (analogous to a kite) to increase the hydrodynamic drag force. Please see Supplementary Figure 23 for a demonstration. If the vessel has a comparable size with the head then the pressure gradient forming across the magnetic head (due to partial obstruction) substantially increases the propulsion force. We can navigate μ probes in vessels with comparable width even if there is a very thin layer of fluid between the head and the channel walls. To demonstrate this feature, we released μ probes with 250 μm head into a 300 μm channel (**Figure R9** and **Movie R3**). The structure barely bent under magnetic actuation due to the confined space yet the μ probe successfully navigated through bifurcations. The navigation works at low flow rates down to $0.5 \text{ cm}\cdot\text{s}^{-1}$ due to the increased pressure gradient that is present around the structure. We show these results in the Supplementary Figure S8 and Supplementary Movie S9.

Figure R9. Flow-driven navigation in channels with a size (300 μm) comparable to the diameter of the magnetic head diameter (250 μm) at $\bar{u} = 0.5 \text{ cm}\cdot\text{s}^{-1}$. **a** A μ probe is deployed into a channel branching into daughter vessels with 20° angle. Scale bar, 5 mm. **b** The magnetic head is steered towards the top channel (top) and released to advance further in the top channel under the control of flow (bottom). **c** The same control strategy is applied to steer the μ probe into the bottom daughter vessel.

7. The μ probe is based on harnessing hydrokinetic energy and using magnetic actuation, and conventional approach relies on a finite proximal force, and the contact pressure exerted on the outward wall to transmit forces distally. According to Supplementary Fig. 2, the tip of the μ probe followed almost identical trajectories while it was being pulled forward by the flow or pulled back by the insertion device. However, when pulled back, the conventional approach could release the force and reduce the damage to the vessel, the μ probe might do damage to the inner wall by the force of onward blood flow. The force analysis (including the friction at corners with high curvature) might be needed to convince the safety of μ probe during the guiding process and retraction process as compared to conventional approach.

Authors' Response: This is an excellent remark and we thank the reviewer for emphasizing the importance of safety and potential vessel wall damage during antegrade navigation or retrieval of the μ probe, possibly caused by friction. We performed a series of calculations and experiments to quantify all the forces acting on the vessel walls and evaluate the risks. In summary, the results showed that the forces associated with the navigation of our μ probes are orders of magnitude smaller compared to the forces reported for conventional push-based navigation techniques, ensuring the safety of the presented technology. We have also performed navigation trials in phantoms made out of agarose gel to empirically show that the forces acting on the walls are insufficient to cause damage. **The complete force-damage analysis is incorporated into the maintext through Supplementary Note 5.**

In order to navigate the conventional catheters and guidewires through different arteries in the body, the operator relies on a combination of haptic and visual cues, achieved by sensing the small axial forces and torques at the fingertips combined with fluoroscopy imaging. Catheter navigation is achieved through a combination of insertion, retraction, and twist at the proximal end. Data relating to the forces exerted during conventional endovascular procedures is very limited. We did an extensive literature survey on tool forces applied by operators and contact forces resulting from catheter-tissue interactions and found a very relevant article that reports quantitative values (Rafii-Tari et al., Ann. Biomed. Eng., 2017). In that article, the advancement and retraction forces were measured inside silicone-based, anthropomorphic phantoms using a force-torque (F/T) sensor attached to the proximal end of the catheter. To provide direct measurement of the forces exerted on the vasculature, the phantom was rigidly coupled to a 6-DoF F/T

sensor at a region corresponding to the the distal end of the catheter. The maximum force values depend on the expertise of the operator, ranging from 1 to 4 N for push forces and 0.5 to 3 N for contact forces. These measurements were taken in an abdominal aneurysm model with a tortuous iliac artery having a single bifurcation. The force values will scale up with the increasing number of turns or bifurcations.

Agarose substrates are widely used to mimic brain tissue as they display the same consistency and resistance to mechanical damage as brain parenchyma. The agarose channels are very soft with a Young's modulus of 2.4 kPa (Markert et al., J. Mech. Behav. Biomed., 2013), making it quite challenging for the endovascular devices to navigate without deforming and damaging the walls. We could not detect any damage (perforation, dissection, etc.) on the vessel walls of relatively long and tortuous vasculature during advancement or retraction of our μ probes (**Figure R10a**, **Movie R5**, and **Movie R6**). The μ probe did not even deform the walls at high-curvature turns, confirming the very gentle nature of the μ probe-wall interactions. As a reminder, this was also the case for the ex vivo studies in the rabbit ear (see Fig. 6) where no perforation was observed, even when purposely tried. On the other hand, it was impossible to push a very small (arguably the smallest and most flexible) guidewire (Hybrid 0.007-inch, Balt Extrusion, France) through this tortuous vasculature. Indeed, the guidewire perforated the walls at multiple locations during every trial. Notably, we observed occasional elastic energy release during the advancement of the guidewire, which led to extensive damage (**Figure R10b,c**). Furthermore, the gel was not able to withstand the compression forces of the proximal segment that is required to transmit the push forces distally. An active interventional neuroradiologist with 15 years of experience who has always been working in high volume centers (> 1000 interventions per year) performed the tests.

Figure R10. Navigation inside a very soft agarose phantom to visualize deformation and damage on the walls. Channels were casted from 0.5 w% agarose ($E = 2\text{-}3\text{ kPa}$). **a** Flow-driven μ probe effortlessly and safely completed a track of 13 cm tortuous channel in 1.5 s under $\bar{u} = 25\text{ cm}\cdot\text{s}^{-1}$. **b**. Abrupt release of energy during the pushing of the guidewire navigated through a 0.017 microcatheter, leading to wall perforation. **c** Perforated and damaged gel wall during the advancement of the guidewire due to contact forces. Images shown in (b) and (c) correspond to the dashed rectangles shown in (a). The guidewire was inserted from the right. Scale bars, 5 mm (a, b) and 2 mm (c).

There are three different situations where the μ probe may cause damage to the vessel.

- 1. During advancement.** The flow pulls the μ probe against the vessel and causes perforation. This may potentially happen at corners with high curvature or bifurcations.
- 2. During retraction.** While retracting the μ probe against the flow, friction forces may induce dissection on the inward wall of the turns.
- 3. During steering.** Magnetic torque that is accidentally applied in the wrong direction pushes the magnetic head against the wall and causes perforation.

We showed that the 3rd scenario does not pose any risks in our response to the 5th comment of the first reviewer. Here, we show that the first two cases are also safe.

(1) During advancement, the μ probe is transported by the hydrodynamic drag force, F_D , which applies a distributed tension (pull force) along the entire structure. The normal forces F_N at the corners where the μ probe contacts the wall scales with the relative velocity Δu of the fluid with respect to the μ probe

$$F_N \propto F_D \propto \Delta \bar{u} = \bar{u}_{fluid} - u_{advancement}$$

where \bar{u}_{fluid} is the velocity of the fluid and $u_{advancement}$ is the rate at which the μ probe is released. The tension on the μ probe and the normal forces on the inward wall of the turns is maximized when $u_{advancement} = 0$ and $\bar{u}_{fluid} > 0$. Here, we assume $u_{advancement} = 0$ to take into consideration the highest normal forces that may act on the μ probe.

To calculate the normal forces acting on the vessel walls, we have to quantify the drag forces as a function of μ probe length. To this end, we developed an experimental platform that consists of a straight channel where the μ probe was inserted and exposed to flow with constant velocity (Figure R11a). The μ probe was clamped to an external force sensor (SI-KG7, WPI) that is sensitive enough to report the sub-mN tension on the μ probe, which is solely determined by the fluid forces as the μ probe does not contact the channel walls. The plot shown in Figure R11b reports the total drag force per μ probe length for the following parameters: $\bar{u} = 10 \text{ cm}\cdot\text{s}^{-1}$, μ probe width $h = 200 \text{ }\mu\text{m}$, dynamic viscosity $\mu = 1 \text{ mPa}\cdot\text{s}$, density $\rho = 1000 \text{ kg}\cdot\text{m}^{-3}$. We provided an analytical model in the Supporting Information (SI Note 1) that is based on resistive force theory. Assuming $\text{Re} > 20$, the drag force can be calculated using the following formula:

$$F = 2.7 * (h\rho\mu)^{0.5} * U^{1.5} * l$$

Figure R11. Quantification of the drag force. **a** Illustration showing the experimental setup with the force sensor, advancement mechanism, and the fluidic device. **b** Experimental data and simulation of the total drag force acting on a μ probe subjected to an average flow rate of $\bar{u} = 0.1 \text{ m}\cdot\text{s}^{-1}$ at $\text{Re} > 20$. The μ probe length is the part of the μ probe subjected to viscous stresses. Each experimental measurement was repeated at least three times.

The experimental results are in good agreement with the calculated values and the observed offset between simulations and experimental data (on average $30 \mu\text{N}$) is due to the fact that we ignored the magnetic head in our calculations. After validating the analytical model, we calculated the drag force exerted on a 12 cm-long μ probe by the blood flow ($\mu = 3.5 \text{ mPa}\cdot\text{s}$, $\bar{u} = 10 \text{ cm}\cdot\text{s}^{-1}$, density $\rho = 1060 \text{ kg}\cdot\text{m}^{-3}$). The total drag force is $F_N = 0.28 \text{ mN}$, which is orders of magnitudes lower than forces involved in the advancement of conventional guidewires. In addition, we assumed that all the forces applied by the fluid will be transmitted to the vessel wall, which is very unlikely due to mechanical instabilities.

We next studied the normal forces generated by the μ probe that is being pulled by the flow on the inward wall of a turn. In this configuration, the highest normal forces are

recorded in a single turn channel. As an extreme example, in a 180° turn, the total force acting on the inward wall is equal to the total drag force acting on the part of the μ probe distal to the turn. In general, the normal force can be written as $F_N = F_D \sin \frac{\alpha}{2}$, where α corresponds to the turn angle. The total normal force for $\alpha = 180^\circ$ (see Figure R12) is calculated as 0.66 mN with a $L = 10$ cm-long μ probe in a vessel filled with blood and under a velocity of $20 \text{ cm}\cdot\text{s}^{-1}$. The forces induced on the wall during advancement are again orders of magnitude smaller than forces reported in standard catheterization techniques.

Figure R12. Normal forces generated at a 180° sharp turn due to the drag force acting on the μ probe of length L .

(2) During retraction, a pulling (retraction) force must be exerted at the proximal clamped end, which applies friction and normal forces to the vessel at the contact points. The blood flow counteracts the retraction forces, and therefore, increases the amount of force required to pull the μ probe compared to the case without the flow. We developed a model based on force-balance equations to study the total retraction forces required to pull back the μ probe. We compared the calculations with the measured values of retraction forces and validated the model.

The retraction forces can be estimated by assuming the inward wall of the turns as a series of cylinders around which the filament is wound. The angle of the turn dictates the arc surface where the μ probe induces pressure, hence friction. See Figure R13 for an illustration.

Figure R13. Calculation of tension on a pulley system. **a** Filament is assumed to slide on a circular object of radius R forming a contact arc region with angle θ (left). The dynamic friction coefficient μ decreases the tension forces across the turn from T_2 to T_1 . Infinitesimal representation of the filament with $d\varphi$ as the contact angle, dN the normal force, μdN the dynamic friction and dT as the difference between the tension forces across the turn (right). The reference frame is the normal n and the tangential t directions. **b** Series of turns with contact angle θ_i between the filament and the object.

The force balance along the tangential (t) and normal (n) axis leads to

$$dF_N = T d\varphi$$

$$dT = \mu dF_N$$

where F_N is the normal force, T is the tension on the rope, μ is the dynamic friction, and θ is the total angle between the inward wall of the artery and the μ probe. The two equations can be combined as:

$$\int_{T_1}^{T_2} \frac{dT}{T} = \exp(\mu\theta)$$

The solution of the integral leads to the Euler-Eytelwein formula, or Capstan equation, that expresses the tension drop across each turn:

$$\frac{T_2}{T_1} = \exp(\mu\theta)$$

For a series of n turns, the cumulative tension T_{Tot} (i.e. force required to retract the μ probe) is given by:

$$F_R = T_{Tot} = T_1 e^{\mu \Sigma \theta} = T_1 e^{\mu(\theta_1 + \theta_2 + \dots + \theta_n)}$$

In our experimental setup, the tension T_1 corresponds to the drag force acting on the μ probe before entering the first turn, $F_{D,Baseline}$. Thus,

$$F_R = F_{D,Baseline} e^{\mu \Sigma \theta} = F_{D,Baseline} e^{\mu(\theta_1 + \theta_2 + \dots + \theta_n)}$$

As described above, in a pulley system the highest normal forces are recorded in the first turn of the channels. The total normal force exerted to the wall on the first turn is calculated by integrating the total pressure acting on the contact arc:

$$F_N = R \int_0^\varphi p(\beta) d\beta = \frac{T_1}{\mu} [e^{\mu\theta} - 1] = \frac{F_R}{\mu} [e^{\mu\theta} - 1]$$

To validate the model, we devised a setup where the μ probe was connected to a force sensor and pulled inside a 3D printed phantom with a well-defined geometry (Figure R14a). The channel has 6 turns each with 90° turn angle (Figure R14b). The retraction forces were recorded while the μ probe was pulled at $2 \text{ mm}\cdot\text{s}^{-1}$ under different flow rates.

Figure R14. Measurement of the retraction force. **a** Illustration showing the experimental setup, advancement mechanism, and the fluidic device. **b** The retraction forces (F_R) were measured in a phantom with 6 consecutive 90° turns under fluid flow. The μ probe was initially placed at the end of the turn 6 and pulled back towards the entry point (to the left).

The experimental results are shown in Figure R15. The regions (1) - (6) represent the passage of the distal tip at each turn. The retraction force increases exponentially with the turn number, as expected. We also see an increase in retraction forces with increasing flow velocity due to the increase in drag forces

Figure R15. Retraction forces recorded during the pulling of a μ probe at $2 \text{ mm} \cdot \text{s}^{-1}$ at different flow rates. Regions (1) - (6) indicate the passage of the μ probe through curved parts of the channel.

We then normalized the data with respect to the baseline drag force, $F_{D, \text{Baseline}}$. The normalized data essentially follow the same curve (Figure R16). Thus, the total force required to retract the μ probe can be estimated for a given number of turns and the measured or calculated value of the drag force acting on the μ probe just before the first turn.

Figure R16. The force required to retract the μ probe as a function of the number of turns. The curves are normalized with respect to the baseline drag force. Blue curve represents the exponential fit for $\bar{u} = 20 \text{ cm}\cdot\text{s}^{-1}$.

We next extracted the dynamic friction coefficient from the experimental data using the analytical formula. As the turn angle θ is kept constant at each turn, the total retraction force is calculated as:

$$F_R = F_{D,Baseline} (e^{\mu\theta})^n = F_{D,Baseline} e^{n\mu\theta}$$

We used an exponential fit to the data recorded at $\bar{u} = 20 \text{ cm}\cdot\text{s}^{-1}$ to quantify the exponent of the exponential, which was found as 0.152. The dynamic friction coefficient is then found as:

$$\mu_{20} = \frac{0.152}{\theta} = \frac{0.152}{0.37\pi} = 0.13$$

By repeating this calculation for different flow rates, the friction coefficient corresponding to different flow rates are found to be between 0.122 and 0.143. The difference may be related to a increased lubrication effect that is manifested with higher flow velocities. These values agree with the literature. For plastic-plastic friction, the reported values are $\mu < 0.1$ with lubrication and $\mu > 0.2$ at dry conditions (www.tribology-abc.com/abc/cof.htm).

After this verification step, we can conclude that the total retraction force required to pull back a μ probe that passes through n number of 90° turns, with dynamic friction coefficient $\mu_{20} = 0.13$, can be calculated using the following simple formula:

$$F_R = F_{D,Baseline} e^{\mu_{20} \cdot 0.37 \cdot \pi \cdot n} = F_{D,Baseline} e^{0.152 n}$$

In our phantoms filled with water, the baseline drag force is calculated as $F_{D,Baseline} = 0.021 \text{ mN}$ (see Figure R11). For a μ probe $l = 5 \text{ cm}$, $\bar{u} = 5 \text{ cm}\cdot\text{s}^{-1}$, and 20 turns, the total retraction force is

$$F_R = F_{D,Baseline} e^{0.152 n} = 0.44 \text{ mN}$$

We can calculate the retraction force under physiological conditions (i.e. blood viscosity and vessel friction) for 20 turns with the same μ probe length and flow rate. The dynamic friction coefficient in the vessels is reported as $\mu = 0.013$ (Takashima et al., Tribol. Int., 2007). Under these conditions, the baseline drag force $F_{D,Baseline} = 0.041$ mN and $F_{Retraction} = F_{D,Baseline} e^{0.013 \cdot 0.37\pi \cdot n} = 0.0423$ mN

Despite the increased drag force provided by the blood, the ~ 10 -fold decrease in dynamic friction coefficient leads to reduced forces. Likewise, the maximal normal force F_N and friction force (F_F) can be calculated:

$$F_N = \frac{F_R}{\mu} [e^{\mu\theta} - 1] = \frac{0.0423}{0.013} [e^{0.013 \cdot 0.37\pi} - 1] = 0.0495$$
 mN

$$F_F = \mu F_N = 0.64$$
 μ N

The total retraction force and the normal forces on the inward walls remain in the μ N-range, assuring that no damage would occur during retraction, perforation, dissection, or otherwise.

8. How to realize the direction of external magnetic field? It seems that the direction of μ probe could be guided only by changing the magnitude of the magnetic field. But can the probe be guided by changing the direction of the magnetic field? What's the difference between the two modes? The author needs more explanation.

Authors' Response: We thank the reviewer for raising this point on magnetic control, which requires further clarification. Along with many other more complex strategies (e.g. changing magnitude and orientation simultaneously), our system allows both options for control of torque: tuning the strength of the magnetic field while keeping the orientation constant, and tuning the orientation of the magnetic field while keeping the field strength constant. To be able to navigate through branches, we need to know the orientation of the daughter artery. While navigating inside phantoms, we applied a magnetic field that is perpendicular to the longitudinal axis of the μ probe head (perpendicular to the main artery) and tuned the strength of the magnetic field to guide the tip to the daughter artery. On the other hand, in the ex vivo experiments with the rabbit ear, we applied a constant magnetic field and dynamically changed the direction to align the field along the direction of the chosen daughter artery. We anticipate that the latter strategy is easier to implement in

clinics. Yet, the first technique enables finer control over μ probe motion and lower power consumption because lower magnetic fields are required for the application of the same torque. As the system is highly damped (laminar flow), we do not expect a significant difference in the equilibrium position and orientation of the head of the μ probe regardless of the chosen control strategy but the transient response may differ.

To provide a demonstration, we applied three different magnetic signals to a μ probe under the same flow conditions (**Figure R17**). In the first case, we kept the orientation of the magnetic field constant (perpendicular to the longitudinal axis of the μ probe, along y-axis in the illustration) like we did in Figures 2-4, and gradually increased the magnetic field strength from 0 to 20 mT. In the second case, we kept the magnetic field strength constant at 20 mT and rotated the field. In the third case, we applied a step function where the magnetic field was instantaneously increased to 20 mT along the y-axis. The results showed that in all three cases the μ probe head was rotated to the same steady-state orientation, 42° , but with different angular dynamics. This result is not surprising as the torque acting on the head depends on the cross product of the magnetic dipole of the head and the externally applied field, which not only depends on the field strength but also the angle between these two vectors. As explained in the previous paragraph, larger angle between the magnetic field generates larger torque and faster rotation. On the other hand, the angular displacement was closer to linear in the second case as the head and the field rotated together. Step function led to the most non-linearity in motion. We also observed large fluctuations in the y-position of the head due to the abrupt application of a large torque and associated fluid-structure interactions.

We identified the type of navigation strategy in the revised version of the maintext and provided a clear illustration in Supplementary Figure S12.

Figure R17. Different strategies for steering the μ probe head. **a** Illustration denoting the major parameters, the coordinate axes, flow rate, and angular displacement as a response to the external magnetic field. **b** Magnetic field vector has two components, magnitude $|B|$ and orientation ϕ . For clarity, we show only 2D control, thus only one angle. A variety of control techniques can be developed based on the dynamic modulation of these two components. **c** The angular displacement curve for three different actuation signals. (Top) $|B|$ is slowly increased from 0 to 20 mT while keeping the the magnetic field perperindicular to the longitudinal orientation of the

channel. (Middle) φ is increased from 0 to 90° while maintaining a constant $|B|$. (Bottom) A pulse is applied where $|B|$ was set to 20 mT and φ to 90°.

9. We know that μ probe can realize controlled-lift and controlled-heading in two-dimensional plane in unidirectional magnetic field environment. We wonder whether does the μ probe need more complex magnetic field control (magnetic field with different orientation) or a unidirectional magnetic field control in a three-dimensional environment? The author needs more demonstrations.

Authors' Response: The same control laws that were used in the article (i.e. unidirectional magnetic field with tunable field strength and rotating magnetic field with constant field strength, see our response to the previous comment for the details) can be used for navigation in 3D. Every bifurcation lies on a single plane (**Figure R18a**), and the operator (or the computer) will first tune the orientation of the magnetic field either perpendicular to the main artery or along the direction of the daughter artery. The next step is the application of the magnetic field, either perpendicular to the incident artery at a given field strength (**Figure R18b**) or by gradually increasing the field strength until the orientation of the μ probe head aligns with the direction of the daughter artery (**Figure R18c**). Inside the phantoms and rabbit ear, we essentially had the whole vasculature on a single plane so a single camera view was enough to determine these geometric features. In a clinical setting, either the C-arm of the fluoroscope will be rotated to determine the plane on which the bifurcation lies or a two-view fluoroscope system will be used to determine the 3D orientation of the main and daughter arteries.

As a response to the 5th comment of the reviewer, we demonstrated the capability of navigating in a phantom with 3D vasculature. In that demonstration, we applied magnetic field with a constant field strength and only changed the orientation of the magnetic field according to the orientation of the daughter artery. The operator determined the orientation of the daughter artery using the images taken by two cameras observing the workspace from two different orthogonal sides of the phantom. The technique was sufficient to navigate in the 3D phantom. The operation would have been done much faster if a computer algorithm had made the calculations.

We identified the type of navigation strategy in the revised version of the maintext and provided a clear illustration in Supplementary Figure S12.

Figure R18. 2D projection of proposed 3D navigation control strategies. **a** Large view of a vascular tree example presenting three bifurcation (I, II, III) leading to the target region (black circle). Fluoroscope views are boxed with black dotted lines. **b** Perpendicular actuation control: the magnetic head is rotated with minimal magnetic field by orienting it perpendicular to the incident vessel (assumed parallel to magnetic head). **c** Target vessel direction control: magnetic field is oriented towards the target vessel and the magnetic field increased until the head is properly oriented. Dark grey dotted lines indicate the target trajectory.

10. The author mentioned that the technology enabled the simultaneous deployment of multiple μ probes. Since the guidance depends on the magnetism, when multiple μ probes are introduced, will the magnetic field affect other μ probes already arranged?

Authors' Response: During the guidance of the active μ probe, the orientation of the heads of previously deployed μ probes will temporarily change. However, the position will remain the same as there is no external pulling or pushing forces acting on the μ probe except the constant drag forces. As a rule of thumb, the smaller the vessel, the steadier

the μ probe head. During navigation, magnetic fields are only applied at bifurcations and for a very short period of time. We already showed that the magnetic torque acting on the head is safe even for the worst-case scenario (see our response to the 5th comment of the first reviewer).

11. In Fig.6a, the flow sensing μ probe was captured to show the heater and sensor elements along with the magnetic head. But our readership may not know the which part of photo corresponds to which part of the μ probe. A diagram might be needed to show the corresponding relationship between the head of probe and heater, sensor element.

Authors' Response: We annotated the picture of the μ probe to identify different electrical components (**Figure R19a**). The magnetic head does not play a role in the sensing operation. The generated heat travels from the heater to the sensor at a rate determined by the velocity and the shear of the flow (**Figure R19b**). **Figure R19c** shows the procedure for calibration. The time-of-flight is calculated by subtracting the pulse start (t_1) from the peak time (t_2). The shear stress is calculated from the flow conditions and the channel geometry (see **Supplementary Note 3, 4**). Please refer to our reply to the next comment for further clarification regarding the working principles of thermal mass flow meters.

Figure R19. The design and working principle of the flow sensor. **a** Detailed description of the electrical components of the flow sensor. **b** Illustration showing the heat transfer by convection from the heater towards the sensor that is governed by the fluid flow inside the channel. Two source

meter controls the time at which the heater is activated, t_1 , and records the time at which the maximum temperature is recorded by the sensor, t_2 . **c** Time-of-flight (TOF) is computed by subtracting t_1 from t_2 . The calibration curve is generated by plotting the TOF value with respect to the average flow velocity for a given channel geometry.

12. What are the functions of the heater and sensor? Heating in blood vessels could cause damage on blood vessels, which could mislead our readers. Further clarification might be needed.

Authors' Response: The working principle of the flow sensor is based on the measurement of the change in the resistance of the sensor element (due to an increase in temperature) as a response to the controlled release of heat by the heater element. The heater element is generating heat because of the current passing through a high-resistance serpentine circuit (Joule heating), which allows us to precisely control the temperature injected into the stream. Similarly, the sensor element consists of another serpentine that reports temperature changes by changing its resistance. The heater element is placed upstream with respect to the sensor and it generates a very short (~10 ms) heat pulse (Fig. 6b) that is transported downstream by the fluid flow. Higher flow rates lead to shorter travelling times, which is defined by time-of-flight (TOF). Figure 6b shows the time it takes for the sensor to react to the heat pulse after its generation at different flow rates.

To quantitatively verify that the device does not trigger coagulation or damage the vessel lumen, we built a finite element model of the system and analyzed the temperature around the device. **Figure R20a** illustrates the position of the μ probe inside the channel during the experimental recordings and the region which the computational model is based on. In the numerical model, the shear stress sensor is assumed to contact the vessel wall surface ($y = 0$). The conceptual flow profile is marked with white arrows for a 2 mm width channel and an average flow rate of $\bar{u} = 0.5$ m/s (**Figure R20b**). The simulation results show the interval with the largest heat front (at $t = 39$ ms from the heat pulse start) and highlights the highly local sensing principle of our sensor. Notably, the thermal boundary layer is considerably smaller than the flow boundary layer. To better visualize the temporal heat propagation, we zoomed in the red dotted box and showed sequential images of the fast measure. The time-lapse close-up images shown in **Figure R20c** quantify the millisecond heat front propagating from the heater towards the sensor unit. At $t = 39$ ms

and only 70 μm away from the shear stress sensor, the temperature has basically already reached the ambient temperature. After 150 ms from its release, the heat has almost completely dissipated. The input pulse generated by the heater is displayed in Figure R20d, which is a one-to-one replicate of the input shown in Fig. 6b. The extracted TOF values are plotted with respect to the shear stress (**Figure R20e**).

Overall, the simulations results show that we can freely choose and tune the input temperature to avoid exceeding 40 °C, which has been reported as a safe value (Kim et al., Nat. Mater., 2011). The sensor is based on temporal measurements; thus, low and rapid temperature pulses are compatible with the sensitivity of the sensor and the working principle.

The simulation results are incorporated into the maintext (page 12) and the following figure is shown in the SI as Supplementary Figure S26.

Figure R20. Computational analysis of the flow sensor. **a** The configuration of the μ probe inside the channel. Black dotted box denotes the part of the device and workspace that is modeled using a multiphysics simulation software. **b** A snapshot from the simulation showing the local heat

distribution. The sensor works with low heat generation that leads to a temperature increase of 3°C on the surface of the sensor. Heat is rapidly and locally dissipated. **c** Close-up, time-lapse images from the simulation of the area inside the red box shown in (b). Scale bar, 200 μm . **d** Pulse generated by the heater during the simulations. The corresponding TOF calibration curve is shown in (e).

13. There was no dense bone effect in rabbit ears. However, if the device is applied to the intracranial, the high density of the skull has an impact on the visibility of the μ probe. DSA images of the μ probe in skeletal environment should be needed to show the application value of the device.

Authors' Response: We thank the reviewer for this comment. We used rabbit ears as an ex vivo model system precisely because of the absence of dense tissue, which enables optical visualization using a standard camera. We verified that our μ probes can be visualized under x-ray imaging. Please see our response to the 5th comment of the first reviewer.

The results of fluoroscope imaging is summarized in the new subsection "The translation of the technology to in vivo applications" and shown in Supplementary Figure 13.

14. The animal experiment was performed on ex vivo Rex rabbit ear. Why not use an in vivo rabbit ear? It could be more convictive if femoral artery puncture was performed and then μ probe was applied in vivo. Besides, other arteries could also be selected to show the three-dimensional state after the application of μ probe.

Authors' Response: As we explained in our response to the comment of the first reviewer, moving on to the in vivo trials requires significant technical contributions and resources such as building of a larger electromagnetic control system and adapting the algorithms to work with fluoroscopic images, and will be the focus of our future work. Here, we aim to explain the physics of structure-fluid-magnetics interactions, show that the overall framework is compatible with the clinical settings, demonstrate functionality with microfabricated electronic μ probes and microfluidic tubes, and validate that the navigation strategy as well as the materials are safe.

The reviewer raised an important challenge for the translation of the technology into in vivo research and clinics. In both in vitro and ex vivo experiments, we were perfusing the vessels with a saline solution and the same perfusion system was coupled to the insertion device. This is an unlikely scenario for in vivo tests because mammals have their own cardiovascular pumping system and we should be able to introduce the μ probes through proper cannulation. Furthermore, it is desirable to harness off-the-shelf tools and components such as an IV-bag or standard syringe for injection instead of a peristaltic pump. We adapted our technology with the following complementary features.

(1) We devised a cannulation system for subcutaneous insertion using standard flexible cannulas (**Figure R21**). Here, we use a pressurized IV-bag or a simple syringe to provide the perfusion. Like the previous version of the insertion system shown in Figure 1b, the system comprises of a rigid rod for controlling the deployment of the μ probe, a commercially available Y-shape adapter for introducing the perfusing and a flexible cannula for the injection. In the previous version, the flow was provided by a peristaltic pump. Here, the flow is provided by a classic IV-bag placed 1 m above the phantom. The hydrostatic pressure was sufficient to provide a gentle flow that drags the μ probe into the vessel. Vessels with higher pressures (e.g. carotid artery) require maintaining a constant pressure difference between the blood flow and the IV bag. The perfusion can also be performed manually using a standard 5 mL syringe. Both systems require very low volumes of physiological fluid for the injection (< 1mL).

Figure R21. Flow-driven deployment using off-the-shelf equipment. **a** A standard cannula was connected to the insertion device that is perfused by a standard pressurized IV-bag. **b** Snapshots of a cannulation process of a PDMS phantom. Scale bars, 30 mm and 5 mm (inset).

(2) We developed a hybrid system that combines our technology with commercially available microcatheters and guidewires (**Figure R22**). Here, the μ probe was bonded to the tip of a 300 μ m guidewire (Traxcess, Microvention, CA, USA) that was inserted into a microcatheter (Rebar 027, Medtronic, Ireland). Proximal perfusion through a 5 mL syringe or a pressurized IV-bag was again sufficient to provide the drag forces to keep the μ probe in tension. The advancement of the guidewire inside the microcatheter is the mechanism for the deployment of the μ probe. The main advantage of this upgrade is the compatibility of the technique with standard surgical materials. Therefore, no additional training is

required for the clinician. Femoral cannulation and guidance of catheter to the main cerebral arteries can be performed using conventional techniques. Once the catheter is in place, the μ probe connected to the guidewire can be deployed in the circulation by perfusing the loaded microcatheter using a saline-bag or a syringe. Once the μ probe is inside the vessel, the prevalent blood flow will guarantee constant tension on the μ probe. No extra perfusion will be required for navigation. Less than 1 mL is required to inject and deploy the μ probe from the microcatheter and the cannula insertion system.

Figure R22. Flow-driven deployment using commercially available endovascular microcatheter. **a** A 3 F microcatheter is connected to a perfusion syringe and a pressurized IV-bag. The μ probe ($\varnothing 150 \mu\text{m} \times 2 \text{ mm}$ magnetic head) is glued to the guidewire and pulled into the microcatheter. Upon introduction of perfusion, pushing the guidewire allows the μ probe to exit the microcatheter (right). **b** Demonstration of the hybrid microcatheter- μ probe device. Once microcatheter reach places that it cannot advance (e.g. a high curvature turn), it can deploy the μ probe that can effortlessly advance with the flow and reach the target location. Scale bars, 5 mm.

Overall, a narrow insertion system (i.e. cannula or microcatheter) ensures higher drag stress on the μ probe and lower total perfused volume. This is particularly useful for small

animal in vivo trials. Also, we already showed that the μ probe can effortlessly be inserted in hollow structures with inner diameter comparable to the size of the magnetic head.

These results are covered in the new subsection “The translation of the technology to in vivo applications” of the revised manuscript and the two new figures are added to SI, Supplementary Figure S14 and S15.

15. As we know, the internal surface of the blood vessel is smooth, which is conducive to the sliding of the guide wire and the probe. The 3D printing sacrificial negative mould was fabricated and used as an vessel simulation. So, the inner surface of mould might be described and tested to convince it.

Authors’ Response: The surface of the channels in the resin or elastomer phantoms is expected to be more adhesive than the internal walls of the blood vessels. In other words, the demonstrations shown in the article were performed in adverse conditions. We repeated the navigation experiments inside a more biomimetic phantom that is fabricated from agarose gel. Agarose is a hydrogel and, as expected, the μ probes experienced even lower friction forces thanks to the lubrication of the gel. Please see our response to the 7th comment of this reviewer for the details of this experiment.

16. As the μ probe material is used in the artery, its anticoagulant properties also need to be tested.

Authors’ Response: We tested the blood compatibility of our μ probes following a standard ISO-10993-4 protocol and showed that the μ probe does not induce neither platelet activation nor platelet aggregation, which are crucial steps in the blood clotting process. Please see our response to the 4th comment of the first reviewer for the details of the test.

The anticoagulation study is summarized in the new subsection “The translation of the technology to in vivo applications” and the results are shown in Supplementary Figure 16.

Reviewer #3 (Remarks to the Author):

The paper reports the flexible endovascular microprobes which were navigated by the flow-driven transportation and magnetic steering. They fabricated the microprobes using the laser-machined thin polyimide substrate attached to the magnetic head. The strongest aspect of the manuscript is the characterization of the advancement of the microprobes through the flow-driven and magnetic navigation. The authors also provide some data to suggest the feasibility of the microprobes for flow rate measurements and ex vivo drug injection. However, in my opinion, the idea does not provide any new significant innovations in the endovascular microprobes. Especially, their navigation strategies are limited due to the dimensions and size of the microprobe. Overall, the manuscript remains unclear points that need to be addressed to improve the scientific rigor, and I would recommend this paper to be submitted elsewhere.

We respectfully disagree with the reviewer on the novelty statement. We hope that the arguments presented in this response letter along with the new data will convince the reviewer otherwise. To our knowledge, there is no article in the literature that explores the possibility of using flow for the navigation of microscopic endovascular devices. Notably, the navigation strategy is intentionally tuned to be able to navigate ultraflexible and microscopic probes, which is impossible otherwise. We do not aim to replace existing endovascular solutions. Our technology will address an unmet need to reach very distal arteries, specifically in the brain, for scientific research and clinical therapy. We strived to address the specific comments of the reviewer.

Specific comments:

1. How can authors navigate the probes when the diameter of the target vessel is similar with the diameter of the magnetic head? If so, the size of the target vessel is too small to generate the drag force and hydrodynamic lift due to the low velocity gradient. Furthermore, the magnetic head cannot be rotated because of the confined inner lumen of the vessel.

Authors' Response: We thank the reviewer for raising this important challenge. We can navigate μ probes in vessels where they can fit even if there is a very thin layer of fluid between the head and the channel walls. To demonstrate this feature, we released μ probes with 250 μm head into a 300 μm channel (Figure R23a) at extreme low flow rates ($\bar{u} = 0.5 \text{ cm}\cdot\text{s}^{-1}$). The structure barely bent due to the confined space yet the μ probe successfully navigated through bifurcations (Figure R23b,c). Notably, velocity gradient, and, as a result, shear stress, increase with decreasing vessel size. Thus, the μ probe will experience even higher pulling forces and therefore the flow-driven navigation will be favored. Moreover, pressure gradients arising from the partial occlusion due to the magnetic head will substantially increase the total drag force exerted on the tip, leading to more powerful pulling forces. As a result, tailoring the size of the magnetic head to be comparable with the target peripheral vessels would improve the deployment efficiency. We show the data in the Supplementary Figure S8 and Supplementary Movie S9, and discuss the results in the maintext (page 10).

Figure R23. Flow-driven navigation in channels with a width (300 μm) that is comparable to the diameter of the magnetic head (250 μm) at $\bar{u} = 0.5 \text{ cm}\cdot\text{s}^{-1}$. **a** μ Probe is inserted in the channel and released towards the bifurcation with daughter vessels diverging at 20°. Scale bar, 5 mm. **b** By orienting the magnet head (top) towards the daughter vessel (heading-control), the μ probe is

steered towards the top channel (bottom). **c** Likewise, heading-control can be used to navigate the μ probe in the other daughter artery.

2. Does the microprobe still advance when there are vortices in a fluid in which the flow revolves around a streamline? The vortices could be produced in arterial branches or the blood vessels with the stenosis, which can interrupt the flow-driven navigation of the probes.

Authors' Response: To reiterate on our answer to the previous comment, our technology enables navigation inside small vessels where laminar or quasi-laminar flow is manifested. Vortices that are capable of interrupting flow-driven navigation exist in larger vessels, in which we are not planning to operate with the presented μ probes.

We have already shown in the manuscript that relatively strong vortices, which are highly unlikely to emerge in small human vessels, do not interfere with the μ probe performance. As shown in **Figure 3d** and **Figure 3e** (also see **Supplementary Movie 4**), the μ probe flawlessly travelled through extreme occlusions around which vortices were generated. Shakes on the magnetic head could be noticed, but were far from pushing the μ probe off-track. The μ probe stiffness is finely tuned to intimately engage with the flow stresses, which in turns does not allow the μ probe to erroneously bend backwards in a small channel. As discussed in our response to the 14th comment of the second reviewer, our technology can be combined with standard catheterization methods, providing an option to release the μ probe directly to small vessels.

3. There should be more information in the 'Magnetic actuation of the tip for controlled bending of the microprobe' section. The authors seem to clamp the proximal tip of the microprobe during the characterizations of the advancement and the navigation strategy. However, the magnetic torque could be changed by the distance between the clamped end and the magnetic head. For example, if the probe is inserted further, the deflection angle of the magnet heads and the probe body will be increased in an external magnetic field compared to the microprobes with shorter insertion depth. Furthermore, when the probe contacts with the wall, the magnetic torque also can be changed because the contact point acts as a new clamped end. The author should provide some estimate for such cases.

Authors' Response: The magnetic torque for a given current input is the same everywhere in the workspace because the Helmholtz coil configuration generates homogeneous magnetic fields. In other words, the orientation of the head is expected to be the same regardless of the insertion depth. The μ probe was clamped proximally like any other endovascular device to keep the released part of the device under the control of the user (otherwise the μ probe would be transported downstream and lost to the flow). Only the head is magnetic and the rest of the μ probe may deform due to the rotation of the head. As illustrated in **Figure R24a**, this deformation is actually quite localized. We defined a static and dynamic region, and the deformation due to magnetic torque is only happening in the dynamic region due to the viscous effects of the flow. This length of the dynamic region depends on the filament stiffness and dimensions, flow rate, channel size and magnetic field strength. In all our experiments, the dynamic region was smaller than 15 mm while the μ probe was clamped >70 mm behind the magnetic head, far away from the head to have any impact on the deformation of the μ probe. In other words, any μ probe longer than 15 mm will behave the same way.

We performed an experiment to clearly show the independence of the navigation from the insertion depth (i.e. μ probe's position in the longitudinal axis of the channel). We applied a magnetic field as soon as the μ probe is inserted into the channel and released the μ probe with constant velocity. As shown in **Figure R24b-d**, the pose of the μ probe (vertical position y_c and angle α) is identical independent of the longitudinal position of the head. The shape of the μ probe remained the same while the head was transported across the channel, validating that steering does not depend on insertion depth in a straight channel.

Regarding the last part of the comment, the magnetic torque does not change once upon contacting the wall as the μ probe is not clamped to the wall. The μ probe can freely rotate and slide on the wall. Contacting the wall changes the direction of motion of the head in the transversal axis of the channel (as shown in **Figures 4c-f** of the maintext). Yet, the angular dynamics of the head due to the magnetic torque remains the same, linearly increasing until reaching the maximum deflection angle with respect to a given magnetic field (**Figure R17**).

Figure R24. Influence of the insertion depth on the pose of the μ probe. **a** Illustration of the experimental setup. Linear positioner controls the insertion depth of the μ probe inside a 2 mm^2 square channel under the influence of constant flow and magnetic field. Unlike continuum devices, the viscous stresses maintain the μ probe under tension. The static region is not effected by the magnetic torque while a relatively short distal dynamic region bends upon the rotation of the head.

b Centroid y-position (y_c) of a μ probe moving slowly inside the channel at a forward speed of 2 mm s^{-1} under $B = 5$ mT and $u = 5$ $cm \cdot s^{-1}$. Red and blue data points correspond to the forward and backward motion, respectively. **c** Magnetic head deflection angle α of a μ probe moving slowly inside the channel at a forward speed of 2 mm s^{-1} under $B = 5$ mT and $\bar{u} = 5$ $cm \cdot s^{-1}$. Red and blue data points correspond to the forward and backward motion, respectively. **d** Snapshots of the μ probe during the forward movement. Black dotted inset represents the superposition of 12 images of the dynamic region taken every 2.5 mm. Scale bars, 5 mm and 2 mm (inset).

4. How reliable is the magnetic head? The magnetic head was composed of the mixture of the PDMS and NdFeB microparticles. The NdFeB alone is not biocompatible. Furthermore, the water can be absorbed into the PDMS and cause the corrosion of the NdFeB, which produces cytotoxic byproducts. The author should evaluate the biocompatibility and reliability of the device.

Authors' Response: As the reviewer pointed out, NdFeB microparticles may corrode in physiological solutions and PDMS is not water-tight. Thus, NdFeB particles that are encapsulated inside the elastomer may also corrode in time and produce cytotoxic by products. Reports in the literature addressed this problem by coating the NdFeB microparticles with silica, polyethylene glycol or dextran (Kim et al., Sci. Robot., 2019, Jia JM et al., Nat. Meth., 2017). Following a recent protocol (Kim et al., Sci. Robot., 2019), we coated the surface of magnetic microparticles with silica. We tested corrosion of four different samples (bare NdFeB particles, silica-coated NdFeB particles, NdFeB particles encapsulated inside PDMS structure, and silica-coated NdFeB particles encapsulated inside PDMS structure) in accelerated conditions by submerging the samples in 0.2 mM HCl for 72h. Results showed that NdFeB particles corrodes in the acid solution while silica-coated particles are not affected, as reported elsewhere (**Figure R25a**). Encapsulating silica-coated NdFeB microparticles in elastomer provides an extra chemical and mechanical layer of protection, especially for acute operations (**Figure R25b**). Considering the duration of the intended operations (minutes to hours) and physiological conditions (pH 7), we do not anticipate any issue regarding corrosion with the coated particles.

For the cytotoxicity, silica-coated magnetic particles successfully created a water impermeable shell that avoids corrosion and subsequent creation of toxic byproducts. To further verify that the exposed polymers do not cause cytotoxicity, we tested cell viability cytotoxicity using an MTT assay (**Figure R25c**). The assay showed that cell viability was

not affected after 12 h incubation with the bare kapton film, PDMS magnetic head with silica-coated magnetic particles or assembled μ probe confirming that the is not cytotoxic.

Figure R25. Cytotoxicity assay using silica-coated magnetic microparticles. **a** 72h corrosion assay in 0.2 mM HCl of suspended NdFeB magnetic particles with (right) and without (left) silica coat. **b** 72h corrosion assay in 0.2 mM HCl of suspended magnetic head casted with magnetic particles with (right) and without (left) silica coat. **c** Cytotoxicity was assessed by the MTT assay using Human Embryonic Kidney 293 cells as previously described by several groups. For this experimental setup, cells were exposed during 12 hours to 3 different samples: bare kapton film (K: $0.004 \times 0.2 \times 10 \text{ mm}^3$), PDMS magnetic head with silica-coated magnetic particles (H: $\varnothing 0.3 \text{ mm} \times 1 \text{ mm}$) and assembled μ probe (P: K + H). The percentage of the cell viability was normalized between untreated cells (positive control – 100% viable) and cells treated with 1% triton lysant (negative control – 0% viable).

Minor comments:

1. In figure 2a, what are the radius of the curvature of the second and third turns?

In figure 2b, why did the probe buckle? If the probe advances along with the centerline of the vessel, the probe body cannot be bended until it touches the wall. In figure 2d, please provide the unit and value of the vertical axis.

Authors' Response: The radii of curvature of the second and third turns are 1.2 mm and 1.8 mm, respectively. The μ probe buckled because we turned off the flow in the channel. In the absence of flow, there is no viscous pulling forces acting on the μ probe and the advancement is solely driven by the pushing forces applied at the proximal end. This bending movement is due to the compressive stresses on the μ probe generated by the pushing force. As soon as the μ probe head experiences friction forces, the μ probe cannot propagate the forces properly and buckles. Even if the μ probe can reach the first turn without touching the walls, it will inevitably contact the wall to make the turn, which immediately leads to mechanical instability and cease of advancement. As we discussed in the text, structures with the given geometry and mechanical properties cannot withstand compressive forces. The overall message is that the navigation of the ultraflexible μ probes relies on fluid flow and our μ probe is not subjected to any pushing force during the trials. The graphs shown in Figure 2d are conceptual. The purpose of these graphs is to show the progression of friction forces and propulsive forces with respect to the traveled distance for the conventional catheters and our μ probes.

2. In ex-vivo demonstration, how did you quantify the amount of the delivered drug at the target vessels? For example, in the case of figure 3e, the flow rates of the drug in each vessel branch will be different due to the difference in the flow resistance based on the diameter of the vessel.

Authors' Response: The total injected volume was manually controlled through a 1 mL syringe and was between 100 μ L and 200 μ L. We did not observe any substantial difference in the injection rate between the two daughter arteries despite we agree that there is probably a mismatch in the resistance between the two arterial branches. Nevertheless, the μ catheter total fluidic resistance exceeds the arterial resistance, which results in a similar perfusion rate.

REVIEWERS' COMMENTS

Reviewer #1 (Remarks to the Author):

The authors have done a great job addressing the concerns of the reviewers. I have no further comments or suggestions.

Reviewer #2 (Remarks to the Author):

In the revised manuscript, the important role of μ probes in clinical application was expounded. We can find out that μ probes has potential to enhance the reachability, reduce the risk of iatrogenic damage, increase the speed of robot-assisted interventions, and enables the deployment of multiple leads simultaneously through a standard needle injection and saline perfusion. The operation mode of μ probes was explained in more detail, and the magnetic field direction was explained in detail in the figures and videos, which helps our readers understand the machines of the device.

The author also added sufficient data, including the imaging of μ probes using two different fluoroscopes in realistic skull area environments, the calculation of the influence of retraction force on the blood vessel wall, and blood compatibility assay performed on fresh human blood under high shear stress condition, and so on. We believe these data can supplement this study from multiple dimensions and make this study more convincing.

In conclusion, we recommend the manuscript be accepted to be published in Nature Communications.

Reviewer #3 (Remarks to the Author):

The authors have responded thoughtfully to the previous comments with the new data. The answers are well-supported by the data. The concept is novel and unique. However, the magnet-based navigation still could be fatal risk for the medical application, especially in brain vasculature.

For example, brain magnetic resonance imaging (MRI) with high magnetic field (3 – 7 Tesla) has widely used to accurate visualization of the cerebrovascular lesions in the brain and probe implantation. Authors showed the magnetic navigation of the μ probe in an external magnetic field under 30 millitesla. However, the magnetic head can damage or apply stress to the vessels when the probe is imaged or tracked by the brain MRI with a strong magnetic field (> 3 Tesla). The customized Helmholtz coil that authors used in the magnetic navigation evaluation could not be used during MRI. Furthermore, the reviewer would argue that the probe still can block the target vessel with high curvature (90o or 180o sharp turn) and having similar size to the magnet head. Authors only showed the navigation in daughter vessels diverging at 20o.

In summary, although this is an innovative and meaningful work, there are still critical issues in in vivo applications. Therefore, the reviewer would like to ask the authors to address the issues described above before being considered for possible publication in Nature Communication.

REVIEWER COMMENTS

Reviewer #1 (Remarks to the Author):

The authors have done a great job addressing the concerns of the reviewers. I have no further comments or suggestions.

We are glad to hear that the reviewer agreed on the completeness of the revision process. We appreciate all the constructive criticism and productive suggestions of the reviewer.

Reviewer #2 (Remarks to the Author):

In the revised manuscript, the important role of μ probes in clinical application was expounded. We can find out that μ probes has potential to enhance the reachability, reduce the risk of iatrogenic damage, increase the speed of robot-assisted interventions, and enables the deployment of multiple leads simultaneously through a standard needle injection and saline perfusion. The operation mode of μ probes was explained in more detail, and the magnetic field direction was explained in detail in the figures and videos, which helps our readers understand the machines of the device.

The author also added sufficient data, including the imaging of μ probes using two different fluoroscopes in realistic skull area environments, the calculation of the influence of retraction force on the blood vessel wall, and blood compatibility assay performed on fresh human blood under high shear stress condition, and so on. We believe these data can supplement this study from multiple dimensions and make this study more convincing.

In conclusion, we recommend the manuscript be accepted to be published in Nature Communications.

We agree that the manuscript became stronger as a result of the review process. We appreciate all the constructive criticism and productive suggestions of the reviewer.

Reviewer #3 (Remarks to the Author):

The authors have responded thoughtfully to the previous comments with the new data. The answers are well-supported by the data. The concept is novel and unique.

We are glad the reviewer changed his mind on the novel aspects of our work.

Specific Comments:

1. However, the magnet-based navigation still could be fatal risk for the medical application, especially in brain vasculature. For example, brain magnetic resonance imaging (MRI) with high magnetic field (3 – 7 Tesla) has widely used to accurate visualization of the cerebrovascular lesions in the brain and probe implantation. Authors showed the magnetic navigation of the μ probe in an external magnetic field under 30 millitesla. However, the magnetic head can damage or apply stress to the vessels when the probe is imaged or tracked by the brain MRI with a strong magnetic field (> 3 Tesla). The customized Helmholtz coil that authors used in the magnetic navigation evaluation could not be used during MRI.

Authors' Response: It is true that our technology cannot be used inside an MRI machine. Thus, it would be dangerous to use our technology, in its current form, along with high-field (> 3 T) intraoperative MRI (iMRI). On the other hand, real-time projection X-ray imaging (fluoroscopy) is the clinical gold standard and widely used during endovascular interventions. It is common practice to take detailed MRI and computed tomography (CT) scans of patients before the surgery, which serve as a map for the navigation of the catheters and detection of diseased tissues (e.g. tumors, arteriovenous malformations, aneurysms). These imaging modalities are high-resolution but slow for real-time imaging. The surgeon needs continuous access to the patient, which is quite challenging to maintain inside a conventional MRI machine. On the other hand, angiographic fluoroscopes with C-arm configuration are convenient and modular, and provide real-time 3D information on the topography of the vessels and the position of the endovascular devices. Once the operation is completed, the patient (or animal for basic research) will be scanned once more for detailed evaluation of the success of the operation. Once again, there is no risk of taking MRI scans as the device will be removed right after the completion of the intervention or recording.

2. Furthermore, the reviewer would argue that the probe still can block the target vessel with high curvature (90o or 180o sharp turn) and having similar size to the magnet head. Authors only showed the navigation in daughter vessels diverging at 20o.

Authors' Response: Following the reviewer's suggestion, we performed an additional experiment where a probe with a 150 μm diameter magnetic head was released into a channel that starts at 300 μm and goes down to 180 μm at the corners. As expected, the device had no difficulty in making sharp turns (angle larger than 180° and curvature radius of 1 mm^{-1}) even at fast release rates (Figure R1). Despite the high-aspect ratio head and its comparable size with the channel, the advancement could be completed in less than 15 ms. The increased fluidic stress conditions occurring at smaller scales, facilitated the transport and redirection of the entire structure through the tortuous channel. We understand the concern of the reviewer but we can assure that we can navigate our probes channels with arbitrary complexity where the head can fit, provided that there is fluid flow that drags the device.

Figure R1. **Advancement in a channel with high curvature and narrow turns.** Snapshots of the advancement of a μ -probe with a magnetic head of 150 μm in diameter and 2 mm in length. The microfluidic channel was forged from a glass capillary (inner lumen $0.4 \times 0.4\text{ mm}^2$) using a Bunsen burner. Scale bar, 2 mm.